# Chromosomal Mcm2-7 distribution and the genome replication program in species from yeast to humans

**Eric J. Foss**[1], **Smitha Sripathy**[1], **Tonibelle Gatbonton-Schwager**[1],
**Hyunchang Kwak**[1], **Adam H. Thiesen**[1], **Uyen Lao**[1], **Antonio Bedalov**[1,2]*

**1** Clinical Research Division, Fred Hutchinson Cancer Research Center, Seattle, Washington, United States of America, **2** Department of Medicine, Department of Biochemistry, University of Washington, Seattle Washington, United States of America

* abedalov@fredhutch.org

**Data Availability Statement:** All data are publicly available at the NCBI Gene Expression Omnibus data repository under accession number GSE150800. Numerical data for generating figures

## Abstract

The spatio-temporal program of genome replication across eukaryotes is thought to be driven both by the uneven loading of pre-replication complexes (pre-RCs) across the genome at the onset of S-phase, and by differences in the timing of activation of these complexes during S phase. To determine the degree to which distribution of pre-RC loading alone could account for chromosomal replication patterns, we mapped the binding sites of the Mcm2-7 helicase complex (MCM) in budding yeast, fission yeast, mouse and humans. We observed similar individual MCM double-hexamer (DH) footprints across the species, but notable differences in their distribution: Footprints in budding yeast were more sharply focused compared to the other three organisms, consistent with the relative sequence specificity of replication origins in *S. cerevisiae*. Nonetheless, with some clear exceptions, most notably the inactive X-chromosome, much of the fluctuation in replication timing along the chromosomes in all four organisms reflected uneven chromosomal distribution of pre-replication complexes.

## Author summary

Gene-rich regions of the genome tend to replicate earlier in S phase than do repetitive and other non-genic regions. This may be an evolutionary consequence of the fact that replication later in S phase is associated with higher frequencies of mutation and genome rearrangement. Replication timing along the chromosome is determined by 1) events prior to S-phase that specify the locations where DNA replication can be initiated, referred to as origin licensing; and 2) the timing of activation of these licensed origins during S-phase, referred to as origin firing. To determine the relative importance of these two mechanisms, here we identify both the binding sites and the abundance of a key component of the origin licensing machinery in budding yeast, fission yeast, mice, and humans, namely the replicative helicase complex. We discovered that, with a few notable exceptions, which include the inactive X chromosome in mammals, the program of replication timing can

and the code are available in the Supporting S1 to S12 Source Files.

**Funding:** This work was supported by the National Institute of Health Grants R01-GM117446 to A.B., T32 CA009657 to T.G.S. and by Scientific Computing Infrastructure at Fred Hutch funded by ORIP grant S10OD028685 National Institute of Health web site: https://www.nih.gov The funders had no role in study design, data collection and analysis, decision to publish, or preparation of the manuscript.

**Competing interests:** The authors have declared that no competing interests exist.

be largely explained simply on the basis of origin licensing. Our results support a model for replication timing that emphasizes stochastic firing of origins that have been licensed before S phase begins.

## Introduction

Eukaryotes organize the replication of their genomes according to programs that ensure that certain regions of the genome complete replication earlier in S phase than others [1–5]. Differences in replication timing can have significant consequences, as late replication is associated with higher frequencies of mutation and genome rearrangement [6–11]. The importance of replication timing is underscored by the observation that it is sometimes modulated according to the utility of the region being replicated: For example, some chromosomal regions containing developmentally regulated genes replicate early only during those developmental phases during which they are activated [12–14]. An even more striking example is the X chromosome in female mammals, where the active X chromosome replicates early while the inactive X replicates late [15–17].

The two main mechanisms for cellular control of replication timing are the regulation of (1) loading and (2) activation of the replicative helicase complex, also referred to as origin licensing and firing, respectively: Only sites where the replicative helicase has been loaded during G1 are capable of initiating replication during the subsequent S phase, and only a subset of these loaded helicases are actually activated [3,18]. Furthermore, because there is a limiting supply of the proteins required for helicase activation, not all of the helicases that will be activated can be activated simultaneously, and firing factors are recycled to allow sequential firing of origins [19,20]. Despite the biological importance of replication timing, the relative importance of helicase loading versus helicase activation in its control has not been established.

The replicative helicase consists of six highly related subunits, Mcm2-7, whose N- and C-termini align to form a barrel-shaped structure with a central channel [21,22]. This hexamer is targeted to DNA at sites bound by the six-subunit Origin Recognition Complex (ORC), encoded by ORC1-6 [18]. The sequences to which ORC binds differ between organisms: In the budding yeast *Saccharomyces cerevisiae*, it recognizes a relatively specific 30 base pair sequence that contains the 11 base pair ARS consensus sequence (ACS) [23,24]; in the fission yeast *Schizosaccharomyces pombe*, the Orc4 subunit contains AT hook domains, which target the complex to AT-rich sequences [25,26]; and in metazoans, ORC does not exhibit sequence preference, but instead recognizes some other feature of DNA or chromatin [27,28]. Regardless of these differences in the sites that ORC selects, the actual mechanism of loading of the helicase complex is the same in these organisms: ORC, Cdc6 and Cdt1 sequentially load two helicase complexes, using the energy of ATP to open each ring [29]. This results in an MCM double hexamer (DH) encircling the DNA, oriented with the N-termini facing inward and the DNA entering and leaving through the channels at the C-termini. Activation of the loaded helicases occurs in response to the rising activity of two kinase complexes, DDK and CDK, and results in the two hexamer rings locked shut with single strands of DNA of opposite polarity going through the two central channels [30].

Here we identify, at nucleotide resolution, the sites where replicative helicases have been loaded in budding yeast, fission yeast, mouse and humans. Across these four species, we observed similar MCM DH footprints, whose ~60 bp size corresponds to that of the DNA protected by this complex in cryo-electron microscopy studies [21,22], but notable differences in their distribution: In budding yeast, complexes were present in sharp peaks comprised largely

of single MCM DHs; in fission yeast, corresponding peaks typically contained 4 to 8 DHs, were more disperse, and showed a correlation with AT content. In mouse and humans, MCM complexes were even more disperse. Nonetheless, we found that, with a few notable exceptions, the pattern of replication timing in S phase in all four species is largely predictable simply on the basis of the pattern of helicase loading in the preceding G1. Models for replication timing that emphasize control at the level of helicase loading, with subsequent firing of origins being largely stochastic, have been proposed by several groups on the basis of single-molecule and genome-wide analyses [31–40]. Our observation that much of the replication timing program can be derived from the pattern of MCM loading prior to S phase lends powerful support to this pleasingly simple and tractable view of replication timing in eukaryotes.

## Results

### Mcm Chromatin Endogenous Cleavage (ChEC) identifies single MCM DH footprints adjacent to ACS at *S. cerevisiae* origins

The MCM complex encircles approximately 60 base pairs (bp) of DNA as a DH whose monomer components are juxtaposed head-to-head at their N-termini with C-termini facing away from each other [21]. To exploit this arrangement of Mcm2-7 hexamers as a way to identify their exact loading sites *in vivo*, we tagged Mcm2, Mcm4 and Mcm6 subunits at their C-termini in *Saccharomyces cerevisiae* with micrococcal nuclease (MNase) [41,42], permeabilized G1-arrested cells, activated the MNase by adding calcium, and prepared libraries for paired-end sequencing using total extracted DNA without any size fractionation [43]. Strains with tagged Mcm proteins exhibited growth rates and cell cycle distribution comparable to those in wild type, indicating that the presence of the tag did not grossly perturb protein function under our experimental conditions (S1 Fig). Because PCR involved in library preparation preferentially amplifies short fragments, which in turn are generated by MNase activity, we predicted that resulting libraries would reflect the sites at which Mcm2-7 DHs had been loaded; in other words, fragments generated by cutting on either side of one MCM helicase complex should be much shorter than inter-origin fragments, and therefore should be preferentially amplified. Furthermore, these sites should be congruent for the three different libraries and correspond, at least partially, to known replication origins. Consistent with this expectation, our sequencing results for libraries prepared from the three different tagged subunits were focused in sharp peaks that coincided with each other (Fig 1A; r = 0.96 and 0.94 for pairwise comparisons between Mcm2-ChEC vs. Mcm4- and Mcm6-ChEC, respectively; S2 Fig) and with known replication origins and replication initiation sites (Fig 2A, left column, middle row). The Mcm2-ChEC signal correlated well with Mcm-ChIP signal from published datasets (r = 0.75 and 0.70) [44,45] but not with the signal obtained with free MNase (r = 0.09; S2 Fig). Furthermore, fragment size distributions for all three libraries peaked in the 50–62 bp size range, consistent with the 60–62 bp protected by Mcm2-7 DHs observed in cryo-electron microscopy studies [21,22] (S3 Fig). We interpret fragment peaks in the 150–200 bp size range as reflections of cleavage by the MNase-tagged protein between flanking nucleosomes and note that such fragments have been observed for other MNase-tagged proteins, including transcription factors [43].

In this report, we visualize individual MCM footprints as heat maps, with fragment size plotted according to genomic location, and relative read depths within each image represented by color intensity [43]. (A detailed description with code showing generation of such a plot from fastq sequence files is provided in the S1 Source File for Fig 1B.) A typical example is shown in Fig 1B: The Mcm2 footprint at the highly efficient ARS1103 was composed of reads in the 50–70 bp size range located immediately adjacent to the ARS consensus sequence

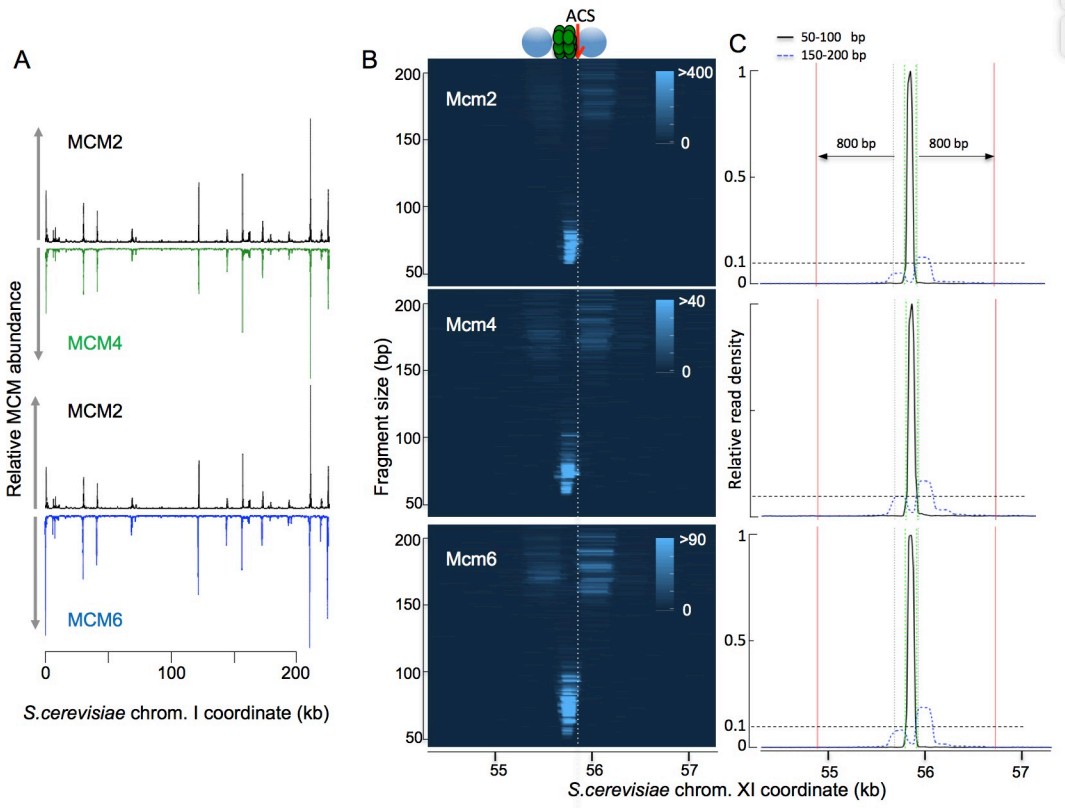

**Fig 1. Tagging of MCM subunits with MNase in yeast reveals MCM helicase binding sites.** (A) Plot of read depths across chromosome I, showing that binding sites for Mcm2 (black), Mcm4 (green) and Mcm6 (blue) are largely congruent. The y axes are scaled to the maxima of the read depths in the three samples, which are 38108, 2502 and 5482 for MCM2, MCM4 and MCM6, respectively, and include all the read sizes. (B) MCM binding sites represented as heat maps that are reminiscent of agarose gels. Chromosomal location is indicated on the X-axis, fragment size is indicated on the Y-axis and per-base pair read depths are indicated by color intensity. (The 3001 x 500 matrices of per-base pair read depths from which these heat maps were generated are available in the S1 Source Files associated with Fig 1B.) All three subunits generate similar footprints. Shown is ARS1103 on chromosome XI. (C) Quantitation of read depths in Fig 1B for fragments in the 50–100 bp (solid black) and 150–200 bp (dotted blue) ranges, which capture signal from MCM DHs and nucleosomes, respectively. The y axes are linear starting with 0 and are scaled such that the maximum read depth is 1.Vertical dotted green lines mark 60 bp from each side of the peak of MCM signal; vertical dotted black lines mark the origin boundaries, as defined in SGD; vertical solid red lines mark 800 bp on each side of the origin boundaries. We use the ratio of MCM signal within the green lines divided by MCM signal within the red lines as a metric for assessing the fraction of MCM signal that arises from a single DH, as described in the text. Source Files for generating the figures are provided as a supplement.

(ACS), and was flanked on both sides by 150–200 bp reads that correspond to nucleosomes. MNase-tagged Mcm4 and Mcm6 generated similar footprints (Fig 1B and 1C), supporting the notion that footprints reflect binding of the entire MCM complex. The distribution of both MCM- and nucleosome-sized particles at this origin, which we define operationally as signal arising from sequencing library fragments that are in the 51–100 bp and 151–200 bp ranges, respectively, are shown in Fig 1C. The concentration of the MCM signal within 60 bp of the MCM peak (marked by green lines), coupled with virtual absence of an MCM signal in the surrounding 1.5 kb of DNA rules out the presence of multiple MCM complexes at this origin and suggests that licensing of ARS1103 involves, predominantly or exclusively, loading of exactly one MCM DH. To determine the speed and efficiency of Mcm2-MNase digestion and to estimate the fraction of origins at which MCM had been loaded, we analyzed a time course of MNase digestion at ARS1103 by Southern blot: DNA was extracted from permeabilized cells

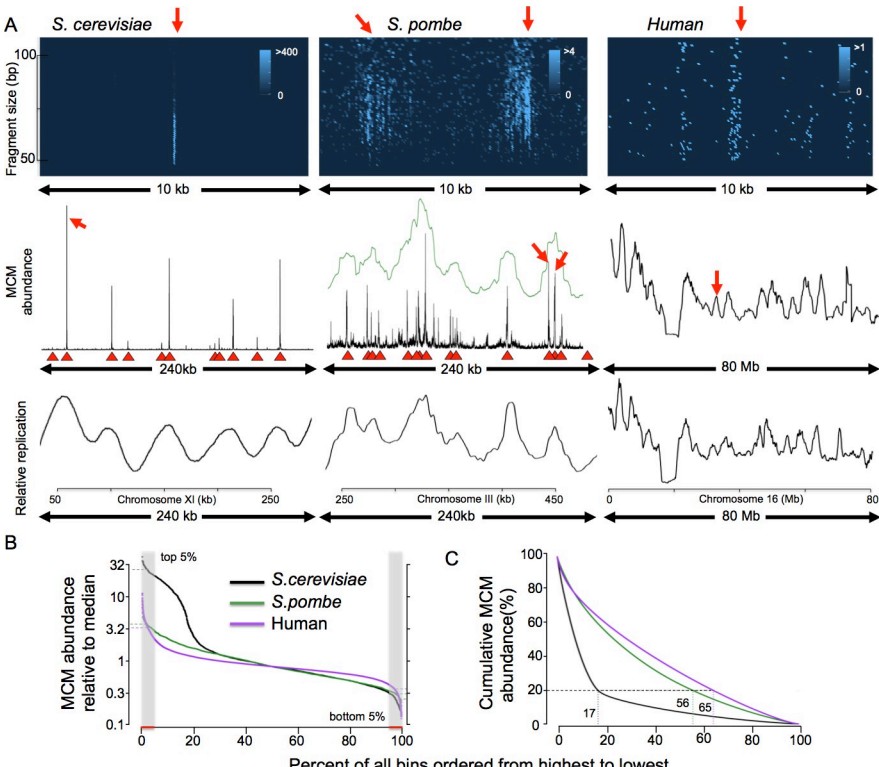

**Fig 2. MCM distribution and replication in yeasts and human.** (A) 3x3 grid summarizing MCM binding and DNA replication in *S. cerevisiae*, *S. pombe* and human (HeLa). Left 3 panels show budding yeast (chr. XI), middle three panels show fission yeast (chr. II) and right 3 panels show HeLa cells (chr. 18). For Mcm2-ChEC experiments (top 2 rows), budding yeast were arrested for 2 hours with alpha factor, log phase fission yeast were transferred to medium containing 15 mM HU for 3 hours, and HeLa cells were arrested with contact inhibition. Top row: Mcm2 binding visualized by plotting chromosomal location on the x axis, fragment size on the y axis, and read depth indicated by color intensity, as described in the text. The budding yeast image corresponds to ARS1103 and the fission yeast to II-448. Middle row: Mcm2-ChEC read depths of all fragment sizes are plotted across 240 kb (both yeasts) or 80 Mb (HeLa). Green line in the middle plot (*S. pombe*) shows Mcm2-ChEC read depths divided into 1 kb bins and smoothed with a 15 bin sliding window. Red triangles in both yeasts indicate known origins of replication, as listed in oridb.org. Red arrows indicate regions of correspondence between the top and middle rows. Bottom row: Replication as determined by S to G1 ratio of flow sorted cells for budding and fission yeast, and by BrdU incorporation for HeLa. (B) MCM distribution expressed as fraction of median value for 5 kb-binned Mcm2-ChEC signal. *S. cerevisiae*, *S. pombe*, and HeLa are shown in black, green and purple, respectively. For each organism, the ratio of the median value in the 5% of bins with the highest MCM abundance (hatched region on left) to that in 5% of bins with the lowest abundance (hatched region on right) provides a metric for the degree of MCM dispersion in each organism. (C) Cumulative distribution of 5 kb-binned Mcm2-ChEC signal of all fragment sizes in *S. cerevisiae*, *S. pombe* and HeLa. The highly focused MCM signal in *S. cerevisiae*, as opposed to *S. pombe* and humans, is clear based on the observation that 80% of the *S. cerevisiae* signal is concentrated in just 17% of the genomic bins, whereas 56% and 65% of the bins were required to capture the same fraction of total signal in *S. pombe* and humans, respectively.

carrying Mcm2-MNase after calcium treatment, digested with Xmn1, which releases a 4.3 kb genomic fragment, and probed with a 261 base pair 3' terminal probe extending up to 17 base pairs to the right of the ACS in ARS1103, as depicted in Fig 3A. We observed a rapid reduction in the abundance of the intact 4.3 bp fragment (~40% reduction within 15 seconds) which proceeded to up to ~90% completion at 5 minutes and without major further reduction at 10 minutes in calcium (Fig 3A and 3B). Loss of abundance of 4.3 kb fragment was coupled with the appearance of an approximately 260 bp fragment corresponding to the fragment released by the MNase digestion between the footprints of MCM and right nucleosome. Similar dynamics of MNase digestion was obtained with tagged Mcm2, Mcm4 and Mcm6 using qPCR (Fig 3B)

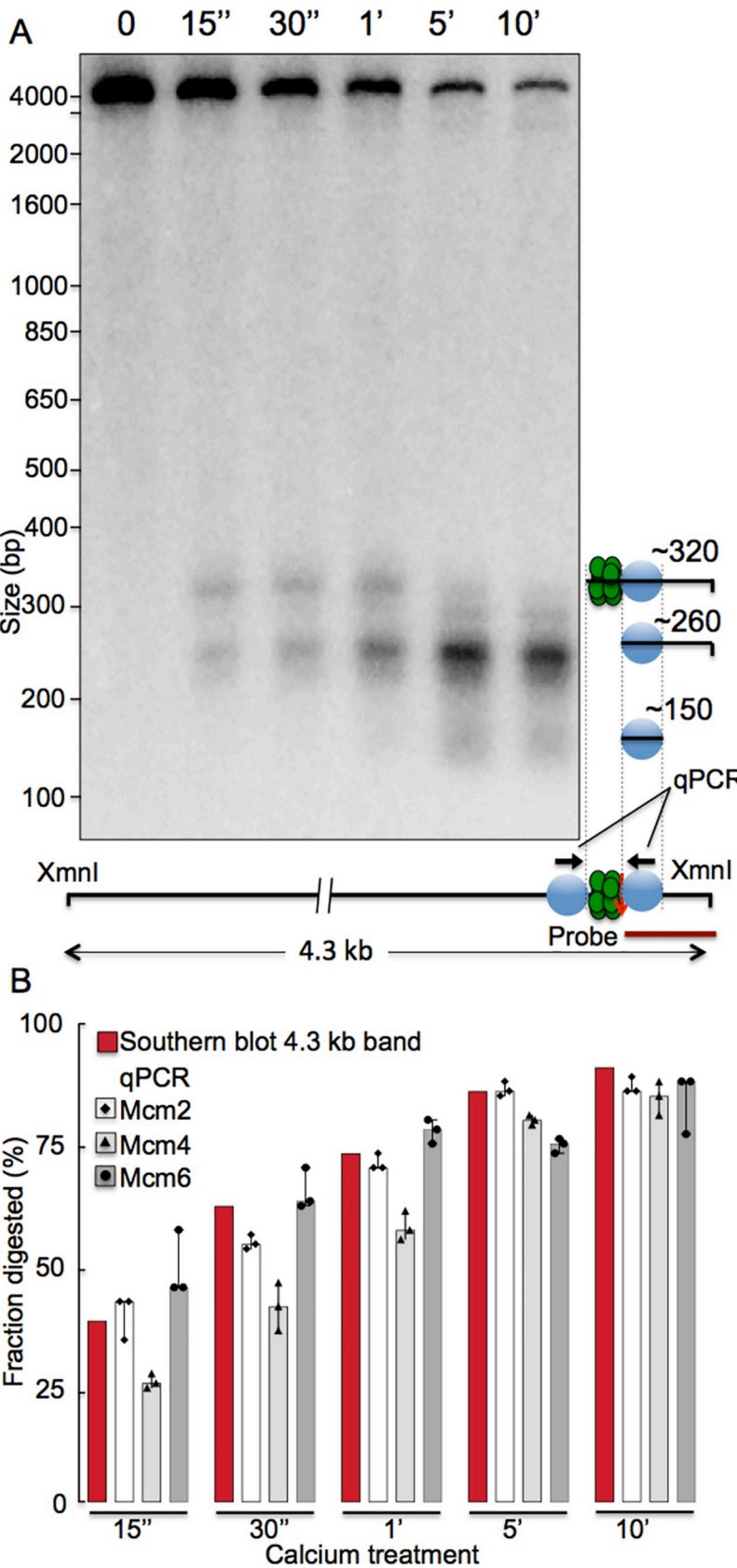

**Fig 3. Kinetics of MNase cutting by Mcm-MNase fusion proteins in budding yeast, as assessed by Southern blot and qPCR.** A. *S. cerevisiae* cells whose endogenous copy of MCM2 was fused to MNase were permeabilized and incubated with calcium to activate the MNase for various times (listed at top). DNA was then isolated, cut with XmnI, and analyzed by Southern blot, using a radioactive probe that hybridizes immediately to the right of the ACS in ARS1103, as indicated in the lower right. Cutting by MNase can be monitored by disappearance of the 4.3 kb XmnI fragment, as it is cleaved by MNase, and by concurrent appearance of shorter bands, whose identities are indicated in the diagram on the right. Green circles depict the MCM complex and light blue circles depict the nucleosomes that flank ARS1103. B. Yeast cells carrying Mcm2, Mcm4- or Mcm6-ChEC constructs at the endogenous loci were permeabilized, the MNase was activated for various times by incubation with calcium, as described above, and cutting at ARS1103 was monitored by qPCR, using primers that flank ARS1103, as depicted by arrows on the lower right. Bars indicate disappearance of the PCR product due to cutting by MNase, with disappearance of the 4.3 kb XmnI fragment in the Southern blot shown for comparison with the results in panel A.

using primers that span the 194 bp region between the footprints of the left and the right nucleosomes as depicted in Fig 3A. These results demonstrate that at ARS1103, Mmc2-7 has been loaded in at least 90% of the cells, and, furthermore, that these loaded origins contain a single MCM DH.

To assess how typical it is for origins to be licensed by the loading of a single MCM DH, we analyzed the distribution of MCM footprints at 343 of the 352 replication origins listed in SGD (9 origins were excluded due to issues of mapping ambiguity). For each origin, we calculated the ratio of (a) the signal arising from library fragments in the 51–100 bp range that falls within 60 bp of the peak of that signal divided by (b) the total signal in the region that is within 800 bp of the origin, indicated with red lines, using 10% of the height of the peak as the cutoff between "signal" and "noise". Fig 4A and 4B illustrates this approach for 12 consecutive origins in a 0.5 Mb region on the right arm of chromosome IV, and data for all 343 origins are in the supplemental information (S4 Fig, S1 Table and the S1, S5 and S6 Source Files for each of the figures and supplemental figures). Mcm4- and Mcm6-MNase, but not free MNase, yielded footprints similar to Mcm2-MNase (S5 Fig). We found that in roughly half of all origins (167 of 343), 80% the signal is concentrated within 60 bp from the peak. We conclude that approximately half of the origins reported in SGD exhibit a single MCM footprint. We note that, although a single MCM footprint at one origin indicates that any individual cell contains either zero or one MCM DH, origins at which we detect more than one MCM footprint do not necessarily reflect the presence of multiple DHs in the same cell; see S6 Fig.

We next compared MCM signal abundance at 165 single-MCM-footprint origins with the 176 origins that contain multiple MCM footprints. (We excluded the two single-MCM footprint origins at the rDNA.) We found that both median and average MCM abundance across origins with single MCM footprints is higher than across origins with multiple MCM footprints (P<0.01; Fig 5A). Cumulatively, single-footprint origins, excluding rDNA, comprise 62% of the total MCM signal. The distribution of MCM signal for single vs. multiple footprint origins (Fig 5B) demonstrates that, despite containing fewer MCM binding sites, origins with single MCM footprints are skewed toward high abundance origins compared to origins with multiple MCM footprints; in other words, origins with a single MCM footprints display higher MCM occupancy. Consistent with higher MCM occupancy, origin activity at single-MCM-footprint-origins was higher compared to multiple-MCM-footprint-origins, as measured by the presence of single-stranded DNA (Fig 5C) [46]. We conclude that replication origins do not require multiple MCM complexes in order to be highly efficient, and that variations in the MCM signal that we observe across the genome predominantly reflect variation in the fraction of cells in which a single complex has been loaded.

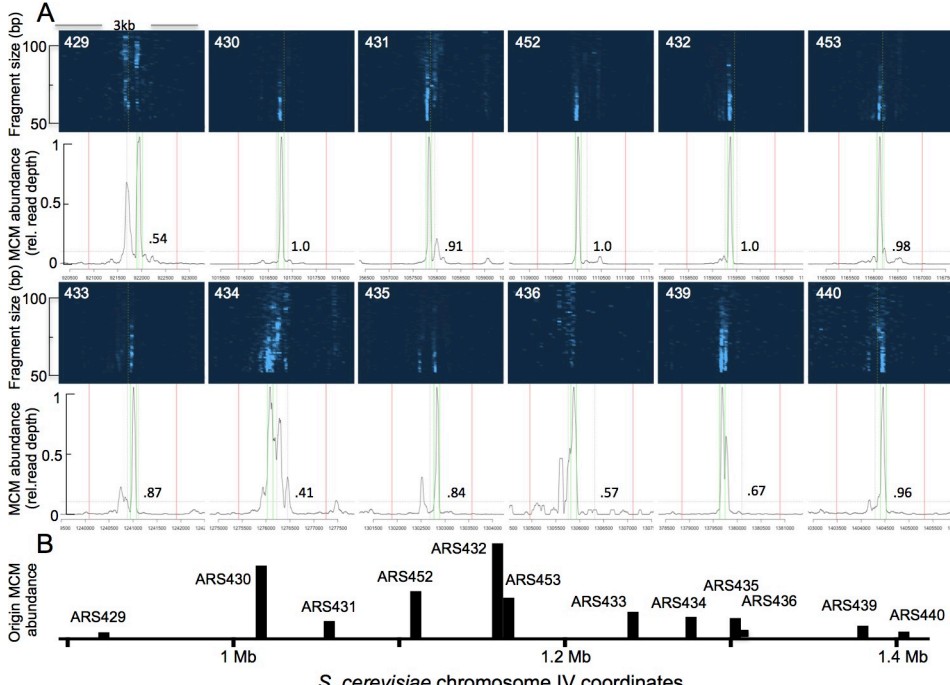

**Fig 4. MCM distribution at 12 consecutive origins on *S. cerevisiae* chr. IV.** (A) 3 kb stretches centered on each of the 12 ARSs in a 0.5 Mb region of chrIV. Top panels show genomic position on X axis and fragment length on Y axis, with read depths represented by color intensity. When an ACS sequence is indicated in SGD, it is depicted with a dashed vertical yellow line. White numbers on blue background indicate ARS numbers, as assigned in SGD. Black traces below blue figures show quantitation of fragments in the 50–100 bp range. Vertical lines mark 60 base pairs on each side of the peak of MCM signal (dotted green), origin boundaries (dotted black) and 800 bp on each side of the origin boundaries (solid red). The fraction of MCM signal arising from a single DH, indicated by the black numbers, is assessed as the fraction of signal within the green lines divided by signal within red lines, using 10% of the height of the MCM peak (horizontal dotted black line) as the cutoff between "signal" and "noise". (B) Chromosomal distribution and relative MCM signal from the 12 origins in A.

## AT-content dictates chromosomal MCM distribution and replication initiation in *S. pombe*

We used the same approach to identify MCM localization in the fission yeast *Schizosaccharomyces pombe*. As in budding yeast, the distribution of fragment lengths was bimodal, with the major peak consistent with the size protected by MCM complexes (S7 Fig). Also like in budding yeast, the Mcm2 peaks largely coincided with previously identified origins of replication (Fig 2A, middle row, middle column). Furthermore, Fig 6A demonstrates a preference of MCM binding for AT-rich sequences, which is consistent with observations that (1) replication in *S. pombe* initiates predominantly from AT-rich sequences [26, 2) fission yeast ORC binds preferentially to AT-rich sequences due to AT hook domains in the Orc4 subunit [25]. However, two features of MCM distribution in fission yeast differed markedly from budding yeast: First, unlike *S. cerevisiae*, where we found a single MCM DH at most origins, we found 4–12 adjacent DH distributed over 500–1500 bp at most known *S. pombe* origins (Fig 2A, top row, middle column; all *S. pombe* origins are shown in S8 Fig). The presence of multiple adjacent footprints at *S. pombe* origins, in contrast to mostly single footprints in *S. cerevisiae*, was observed in both G1- and hydoxyurea (HU)-arrested cells for both species (S9 Fig). Second, we found significantly more minor MCM peaks between origins (Fig 2A, left versus middle column of middle row).

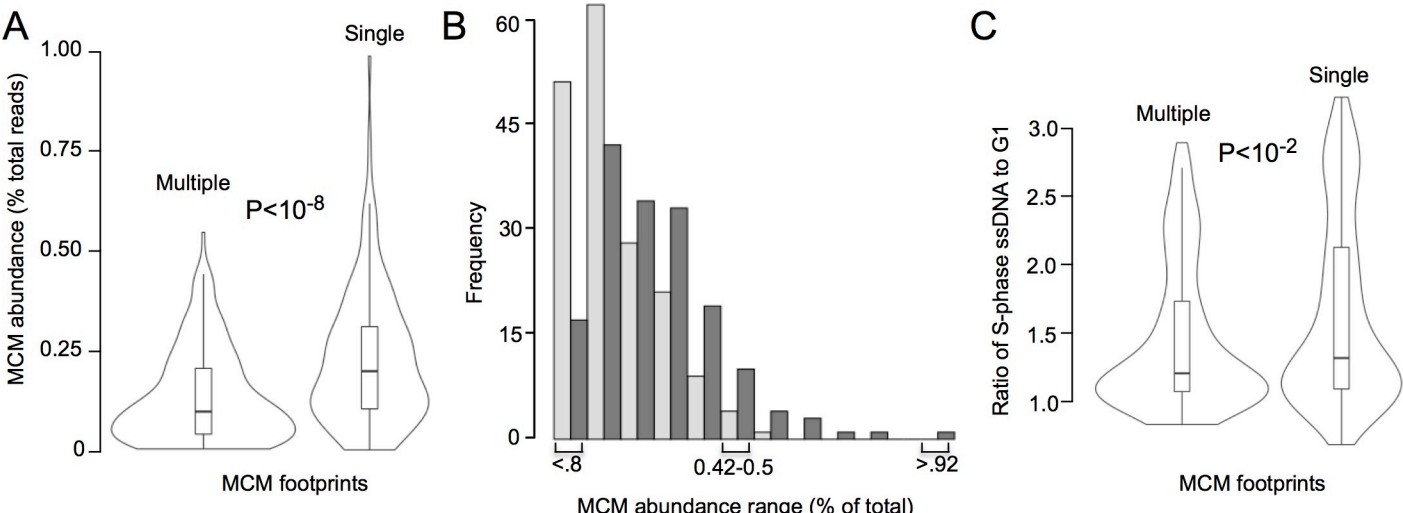

**Fig 5. Comparison of Mcm2 abundance and replication activity at budding yeast origins that have single versus multiple Mcm signals.** A. Total Mcm2-ChEC signal within 800 base pairs of the center of the origin is plotted according to whether at least 80% of that signal originates from within 60 base pairs of the peak of signal (right violin) or not (left violin), as described in the text. All fragment sizes are included in calculating MCM abundance. 341 origins listed in SGD were analyzed, with the two origins at the rDNA excluded. MCM abundance at the 165 origins with a single Mcm2-ChEC footprint was higher than that at the 176 origins with multiple footprints ($p < 10^{-8}$ by two-tailed t test). B. Histograms showing distribution of Mcm2-ChEC signal for origins with single (dark gray) versus multiple (light gray) Mcm2-ChEC footprints. C. Origin activity, as measured by the presence of single-stranded DNA at origins 30 minutes after release of cells into S phase in HU [46], for origins with single versus multiple Mcm2-ChEC footprints ($p = 0.003$ by two-tailed t test).

To determine whether the observed pattern of MCM distribution matches the pattern of replication initiation, we aligned MCM distribution with previously published *S.pombe* DNA polymerase usage sequence analysis (Pu-seq) datasets [47]. Pu-seq relies on a strand-aware method of identifying sites of ribonucleotide incorporation in DNA: By sequencing genomic DNA from strains containing mutant alleles of the leading- and lagging-strand polymerases (pol-epsilon and pol-delta, respectively) that incorporate elevated levels of ribonucleotides, replication initiation zones can be identified as regions of sharp divergence of leading and lagging strand polymerases,, whereas replication termination zones constitute broader regions where divergent polymerases from adjacent initiation zone converge. Examples of initiation and termination zones are shown in Fig 6B, centered on the arrows marked "i.z." and "t.z.", respectively. We observed co-localization between MCM peaks and sites where pol-delta and pol-epsilon diverge, as is shown for a 1 Mb stretch of chr II (Fig 6B) and confirmed in genome-wide comparison of MCM levels at initiation versus termination zones (Fig 6C; $p < 10^{-10}$). Two Mcm2-ChEC replicas were highly correlated (r = 0.86) and exhibited a similar chromosomal distribution pattern (S10A Fig). As expected, initiation zones also showed higher AT content than termination zones (Fig 6D; $p < 10^{-10}$), consistent with the MCM complex's preference for AT-rich sequences. To exclude the possibility that the observed Mcm2-ChEC signal simply reflected MNase preference for digesting AT rich DNA, we repeated the same analysis using DNA extracted from nuclei digested with free exogenously added MNase. Exogenous MNase failed to recreate the chromosomal distribution of the Mcm2-ChEC signal (S10A and S10B Fig), demonstrating that the observed Mcm2-MNase signal indeed reflects MCM distribution along the chromosome. These observations support a model in which AT-content drives chromosomal MCM distribution, which in turn drives the replication initiation pattern in *S. pombe* [37].

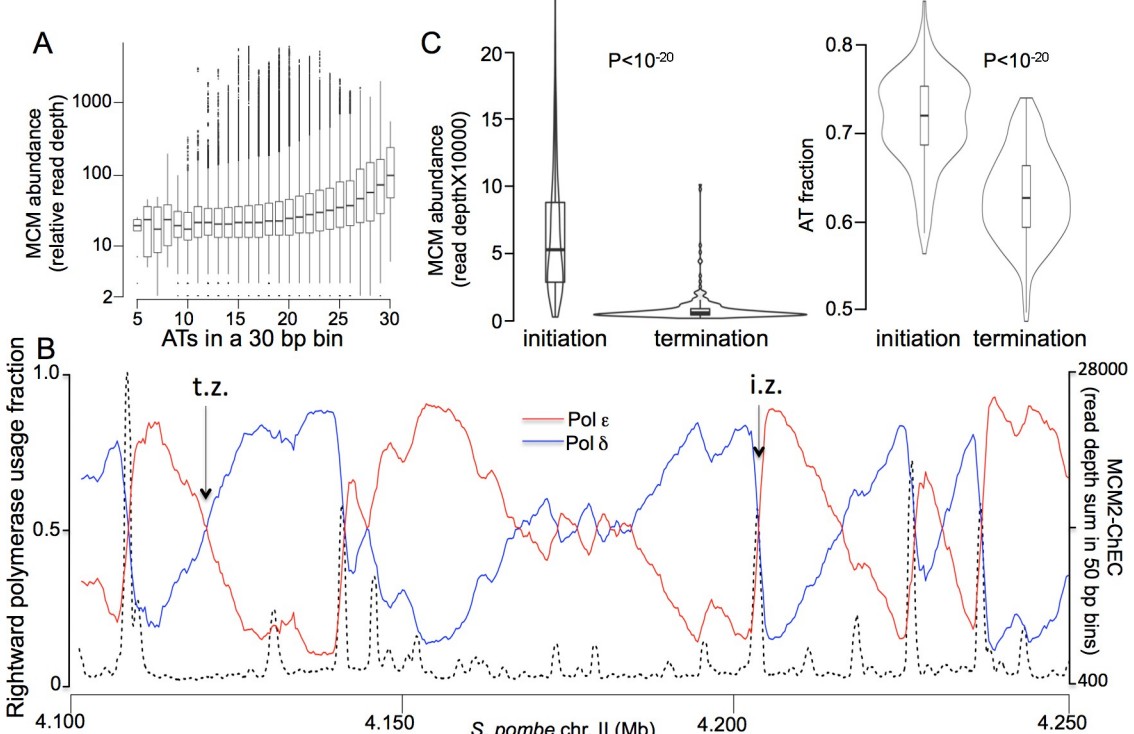

**Fig 6. MCM binding preference in S. pombe with respect to AT content and replication initiation and termination zones.** (A) MCM signal as a function of AT content was determined by dividing the genome into 30 base pair bins and measuring average MCM signal in all bins with 30/30 As and Ts, 29/30 As and Ts, etc. (B) Percentages of rightward moving polymerases (i.e. polymerases that are involved in creating the newly synthesized Watson strand) that are leading- (pol ε) and lagging-strand (pol δ) polymerases, as determined by Pu-seq [47], are shown in red and blue, respectively. MCM abundance, expressed as Mcm2-ChEC signal read depth sum of all fragment sizes in 50 bp bins on the left Y-axis, is indicated by dotted black lines. Zones at which leading- and lagging-strand polymerases diverge sharply indicate replication initiation zones; these appear as sharp downward movements of the blue trace with coincident sharp upwards movements of the red trace. Zones at which leading- and lagging-strand polymerases from adjacent replication initiation zones converge indicate replication termination zones; these appear as gradual downwards movements of the red trace with coincident gradual upwards movements of the blue trace. (C) Comparison of MCM levels at all initiation versus termination zones on chr. II. Initiation and termination zones were determined as described in Materials and Methods. Violin plot shows distribution of relative read depths. Box shows interquartile range; horizontal line shows median; horizontal line in box shows median; vertical lines extending above and below box show distance to point farthest from interquartile range that is still within 1.5 interquartile ranges from the edge of the box; any points outside of this range are shown as individual points. Mean value for initiations was 0.19 (relative per-base-pair read depth), with 95% confidence intervals extending from 0.17 to 0.21; for terminations, mean was 0.023, with 95% confidence intervals extending from 0.020 to 0.027, $P < 10^{-20}$. (D) Comparison of AT content at all initiation versus termination zones on chrII. Mean value for initiations was 0.715 with 95% confidence intervals extending from 0.709 to 0.721, and mean value for terminations was 0.629 with 95% confidence intervals extending from 0.622 to 0.635, $P = 10^{-20}$.

## MCM distribution drives replication initiation pattern of human chromosomes

We next determined the distribution of MCM complexes in human cells using the same technique. To do so, we expressed human Mcm2 tagged with MNase in HeLa cells using a lentiviral vector, choosing an expression level that does not exceed endogenous Mcm2 levels (S11 Fig). As seen in yeast, the distribution of sequenced fragment sizes was bimodal, with the major peak composed of 50–100 bp fragments (S12A and S12B Fig). We observed individual MCM footprints similar to those in both yeast species (Fig 2A, top row, right column), indicating that the basic architecture of the complex is conserved. Two Mcm2-ChEC-seq replicas datasets were highly correlated (R = 0.88) (S13 Fig) but exhibited no correlation with a

previously published free MNase-seq dataset from HeLa cells [48]. The MCM signal distribution along the chromosome in HeLa cells was more disperse than in budding yeast and similar to that in fission yeast (Fig 2A). After partitioning the genome into 5 kb bins and ordering bins based on MCM abundance (Fig 2B), we observed a 9-fold and 11-fold difference in median MCM abundance between the top and bottom 5% of bins in human cells and *S. pombe*, respectively, compared to 94-fold difference in *S. cerevisiae*. The more disperse distribution of DHs in human and *S. pombe* cells was also evident by examination of cumulative MCM distribution (Fig 2C), with the top 65% and 56% of the respective bins constituting 80% of the total MCM, compared to only 17% in budding yeast. These results demonstrate that while the basic architecture of binding of MCM DHs is preserved across species, their distribution in fission yeast and mammals is more dispersed than in budding yeast.

Using published replication timing data, we next asked whether chromosomal regions with higher MCM density were correlated with earlier replication. In a previous study [6], asynchronously growing HeLa cells were given a BrdU pulse and flow-sorted based on DNA content; DNA was then immunoprecipitated with anti-BrdU antibodies and subjected to deep sequencing. Chromosomal distribution of MCM complexes in HeLa cells showed a high correlation with the chromosomal replication pattern, as shown in the middle and bottom panels of the right column in Fig 2A, for the entire 80 Mb of chromosome 18 (r = 0.82). Previous studies have shown that at a finer scale, replication within a single early- or late-replicating region is not uniform; for example, studies using Okazaki fragment sequencing (OK-seq) and BrdU incorporation have shown that replication initiation within these regions is concentrated in the broad zones encompassing from 10 kb and to up 150 kb, that are typically located upstream of transcribed genes, whereas termination occurs within even broader zones [49]. Using OK-seq in HeLa cells, Petryk and al. identified 12,000 initiation and 18,000 termination zones with median and mean sizes of 21 and 25 kb for initiation zones and 24 and 74 kb for termination zones (Olivier Hyrien, personal communication; coordinates in supplemental material associated with S2 and S3 Source Files for Fig 7B). In a fork directionality plot, such as that in Fig 7A, such replication initiation zones are centered at the (from left to right) blue-to-red transition sites, whereas the termination zones are centered at red-to-blue transition sites; such blue-to-red and red-to-blue transitions indicate sites of divergence and convergence, respectively, of leading- and lagging strand polymerases. Just as we saw increased MCM density in multi-megabase early-replicating chromosomal regions (Fig 2A middle and bottom rows of right column), we observed local MCM maxima that coincided with the centers of initiation zones (IZ in Fig 7A), whereas the MCM minima coincided with the centers of termination zones (TZ in Fig 7A), within a single broad region (Figs 7A and S14A–S14F). Compared to termination zones, initiation zones averaged 48% more MCM ChEC signal (95% confidence interval 47% to 50%), suggesting that increased MCM density in these zones underlies the observed replication initiation pattern. The genome-wide significance of the difference in MCM2-ChEC signals at initiation versus termination zones was assessed by unpaired two-sample t-test, with one group consisting of MCM2-ChEC signals in 1 kb bins that overlapped initiation zones and the other group consisting of those that overlapped termination zones (2-tailed p value $< 10^{-20}$). The distribution of the MCM signal along the chromosome using a replicate dataset was similar (S15A Fig). In contrast, analysis of published MNase-seq datasets in HeLa cells treated with free MNase [48] did not recreate the chromosomal distribution pattern observed with Mcm2-ChEC, and nor did it show a significant difference in abundance between initiation and termination zones (S15A and S15B Fig), arguing that our Mcm2-ChEC indeed captures MCM chromosomal distribution and is not merely a reflection of MNase digestion preference.

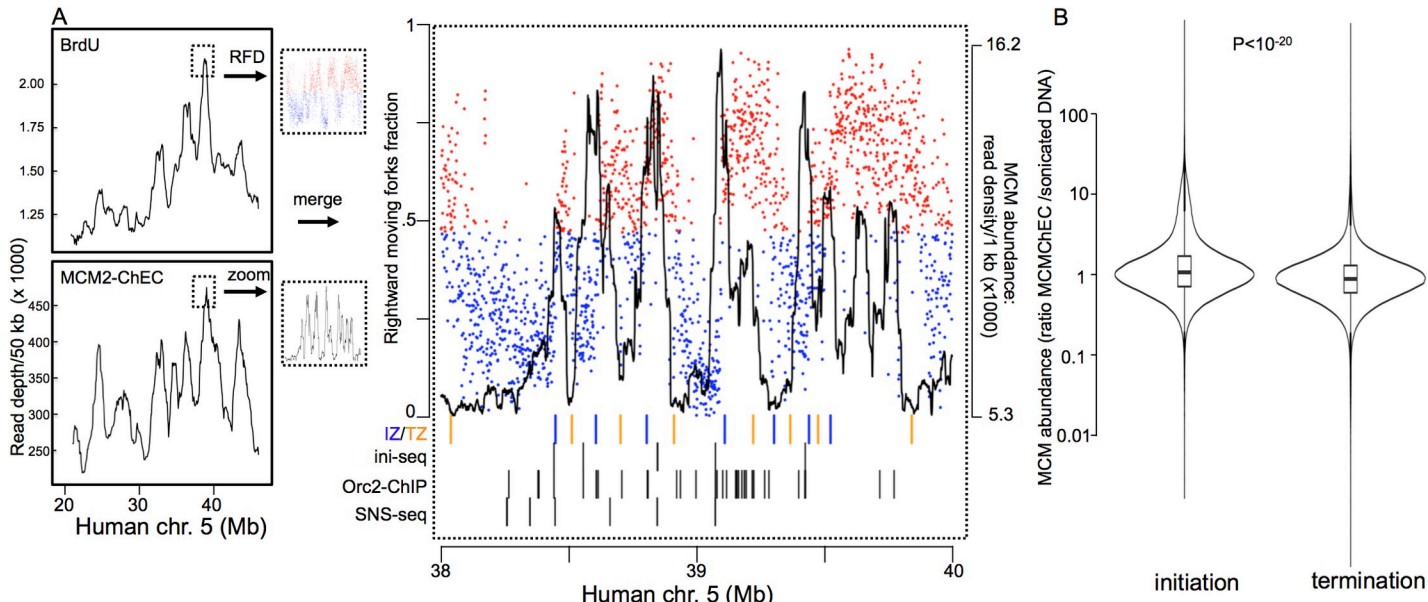

**Fig 7. MCM binding in humans with respect to replication initiation and termination zones.** (A) Replication, as determined by BrdU incorporation [57] upper left panel) and MCM binding expressed as Mcm2-ChEC read depth of all fragment sizes is shown for HeLa cells. Right panel shows blow-up of indicated area with (red and blue dots are used to emphasize direction of majority of synthesis; data from Petryk et al., 2016 [49]) and MCM binding (black). The centers of initiation and terminations zones (IZ and TZ) are marked by red and yellow tick marks, respectively. Black tick marks show locations of Orc-ChIP [50], initiation site sequencing (ini-seq) [51], and small nascent DNA strand sequencing (SNS-seq) peaks [52]. Data from both MCM2-ChEC replicas were included in this analysis; the two replicas are analyzed separately in S15 Fig. (B) Analysis of replication initiation and termination zones on chr18 show higher MCM binding at the former than the latter (P<10−20). MCM binding was expressed as MCM2-ChEC/sonicated G1 DNA read depth ratio. Mean values of MCMChEC/G1 DNA ratios for initiations and terminations were 2.09 (95% confidence intervals extending from 2.08 to 2.11) and 1.41 (95% confidence intervals extending from 1.407 to 1.413), respectively.

The association of elevated Mcm2-ChEC levels with sites of replication initiation was further supported by unpaired two-sample t-tests using data sets for Orc-ChIP [50], initiation site sequencing (ini-seq) [51], and small nascent DNA strand sequencing (SNS-seq) [52]. The two groups used for these t-tests were MCM2-ChEC levels in 1 kb bins that overlapped peaks, using coordinates determined by the authors of the respective studies, and 1 kb bins immediately adjacent to the overlapping bins. Two-tailed p values in all cases were <10e-20. We conclude that at both large (multi megabase) and small (sub 100 kb and sub 10 kb) scales, MCM abundance varies in concert with replication activity.

### Replication modeling in *S. cerevisiae* identifies regions of repetitive DNA that are late replicating despite being well-licensed

While the chromosomal MCM distribution in both *S. pombe* and human cells clearly recapitulates the corresponding replication pattern (Fig 2A, middle and bottom rows of middle and right columns), the discretely focused distribution of MCM binding in *S. cerevisiae* makes the link between chromosomal MCM density and chromosomal replication less apparent (Fig 2A, middle and bottom row of left column; replication data from Yabuki et al. [53]). To determine whether chromosomal MCM density in G1 along the chromosome is sufficient to drive replication timing pattern in budding yeast, we modeled MCM binding and DNA replication *in silico*, assuming that in each cell DNA replication initiates at random from sites bound by MCM in that cell. We reasoned that a computer replication simulations would be particularly helpful because, while we assume that replication timing patterns across large chromosomal regions reflect the integrated activation times of the

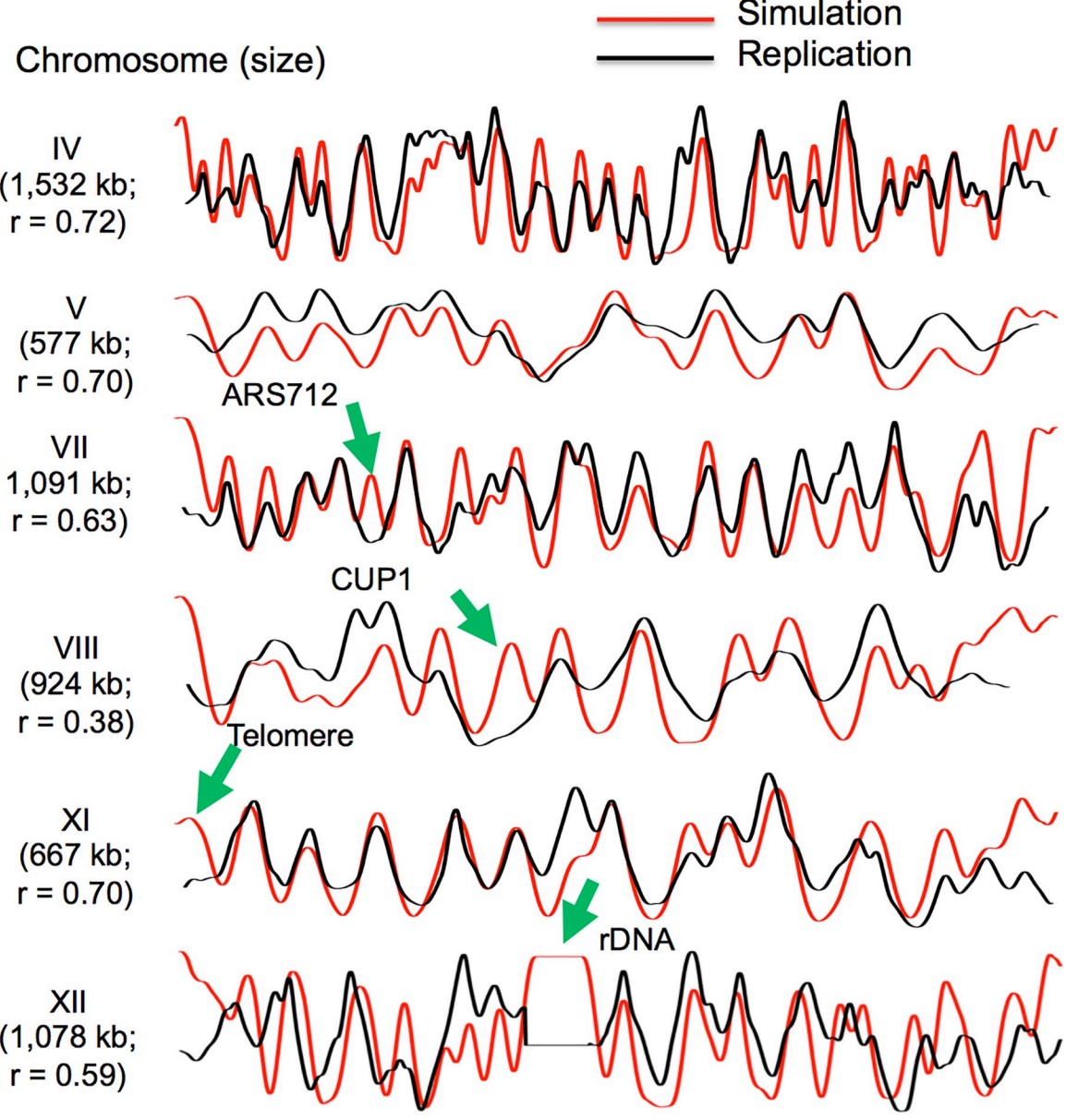

**Fig 8.** Comparison of replication simulations (red) and replication, as assessed by read depth ratios of S phase to G1 phase flow-sorted cells [50] for six *S. cerevisiae* chromosomes. Green arrows indicate sites that replicate later than would be predicted based on simulations including repetitive *rDNA* repeats, telomeres *CUP1* locus and *ARS712*. Delayed activation of these well-licenced origins indicates their control at the level of origin firing. *ARS712*, which has been shown to contain both ORC and MCM in ChIP studies [52], is not active during unperturbed replication, but is active in *rad53* mutants subjected to hydroxyurea [51]. Correlation coefficients (r) between simulations and replication data are indicated. Mcm2-ChEC read depth of all fragment sizes was used for calculating MCM abundance.

individual origins in those regions, the time at which a particular origin is replicated does not necessarily reflect the time at which that origin fired. For example, an origin that is licensed in only a small fraction of cells may complete replication relatively early in S phase due to its passive replication by a nearby origin that is licensed in a large fraction of cells. The stochastic origin firing model, which is similar to those used by others [33, 37, 40], was generally accurate in predicting replication pattern, and it was robust to changes in

additional assumptions about how limitations of origin firing factors restrict the number of origins that can fire simultaneously and how freely those limiting firing factors diffuse (Fig 8, algorithm described in Materials and Methods). However, this analysis also identified chromosomal regions whose replication was not congruent with MCM density. Most notable were repetitive regions, including the *rDNA* and *CUP1* arrays as well as regions near chromosome ends, which replicated later than expected based on our MCM density-driven replication simulations (Fig 8) [54,55]. Several of these well-licensed but late replicating regions in yeast, such as rDNA and telomeres, are subject to regional, gene-independent transcriptional silencing and, as such, represent the equivalent of metazoan heterochromatin. This led us to explore the idea that the late replication of heterochromatin may constitute an important exception to the general rule that replication timing reflects origin licensing.

## Similar MCM abundance, distribution and replication order along early replicating active and late replicating inactive X-chromosome

Two features of the X chromosome in mammals make it ideal for addressing the relationship between replication timing and origin licensing in heterochromatin: First, the late replicating, inactive X-chromosome (Xi) is by far the largest region of mammalian heterochromatin. Second, because the active X-chromosome (Xa) is not heterochromatic, it provides a perfect control, allowing one to separate the effects of sequence from those of chromatin structure on replication timing. To determine whether reduced MCM loading or delayed origin activation underlies late replication of that chromosome, we employed a clonal *M. musculus/M. spretus* hybrid cell line [56]. In these cells, the active X chromosome is derived exclusively from the *M. spretus* parent. Given that the *musculus* and *spretus* genomes contain a single nucleotide polymorphism (SNP) every 50–100 bp, both replication and MCM binding on the two parental chromosomes can be followed independently. To compare MCM distribution and abundance between the two X chromosomes, we expressed mouse Mcm2 tagged with MNase at the C-terminus using a lentiviral vector at a level that does not exceed endogenous protein (S11 Fig) and carried out ChEC. Fragment size distribution was similar to that in yeast and human cells, replicate measurements were highly reproducible, and MCM heatmaps appeared similar to those in HeLa cells (S12B and S16A–S16C Figs). Differences in replication timing between two chromosomes were measured using S-seq, by comparing the S to G1 ratio separately for each allele along the chromosomes, as shown in Fig 9A. We observed only minor allelic differences in replication between autosomes, as previously reported using a similar assay [2], and replication timing of the active *M. spretus* X-chromosome was similar to that of autosomes (Fig 9A upper panels). When we quantified the magnitude of the fluctuations for spretus and musculus alleles along entire individual chromosomes and calculated the ratios of these fluctuations for each chromosome (S17 Fig), the X-chromosome was found to be the single outlier, with a spretus/musculus fluctuation ratio of 2.7 vs 0.98–1.11 (95% confidence intervals) for autosomes, consistent with delayed replication of the inactive musculus X chromosome (Fig 9A, upper right). However, in contrast to their differences in replication timing, we detected no difference in Mcm2 abundance on the inactive and active X chromosomes (Fig 9A, lower right; ratio of *spretus* to *musculus* reads was 1.00 and 1.03 for chromosomes X and 16, respectively). This result, which is consistent with our observation that heterochromatic sequences in *S. cerevisiae* are late replicating despite being well-licensed, demonstrates that delayed replication of the inactive X-chromosome is not a consequence of reduced origin licensing but, rather, of delayed origin activation.

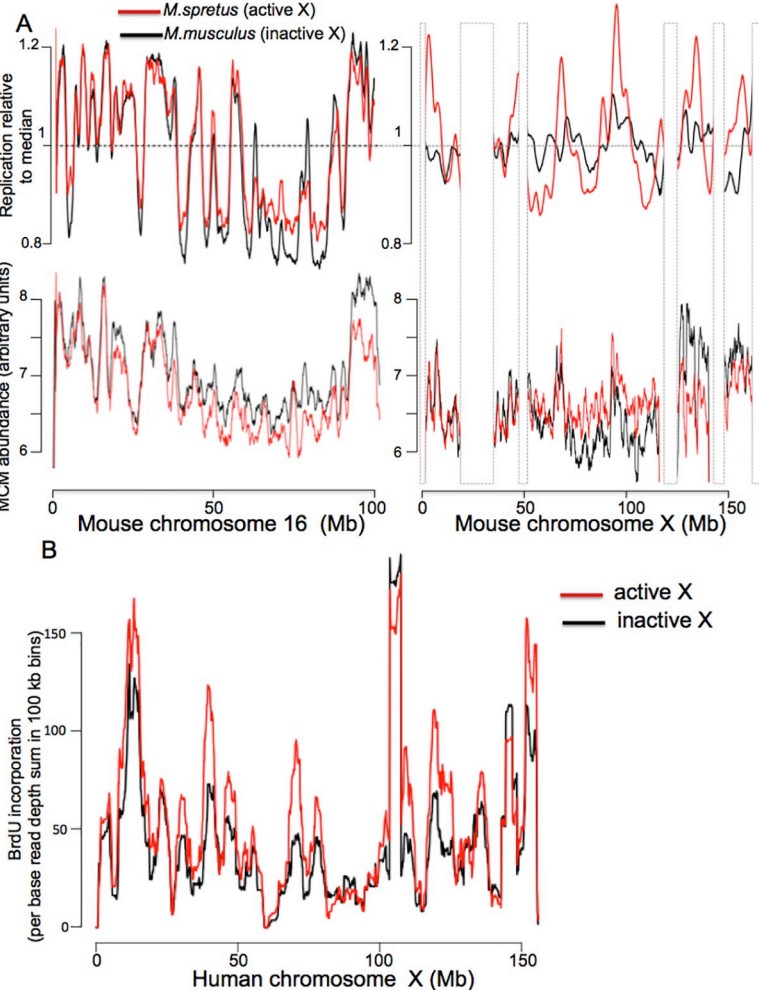

**Fig 9. MCM distribution and replication on the X chromosome.** (A) Replication (upper panel) and MCM binding (lower panel) in mouse Patski cells for chr16 (left) and chrX (right). M. musculus and M. spretus parental chromosomes are shown in black and red, respectively. Replication, measured by S-seq, is plotted as fraction of median read depth, with larger fluctuations from the median reflecting more active replication, separately to each of two chromosomes. MCM abundance is plotted as a ratio of MCM2-ChEC to sonicated G1 DNA read depths using all read sizes, separately for each of two chromosomes as explained in the Materials and Methods section. All Mcm2-ChEC fragment sizes are included in the analysis. (B) Fluctuation in BrdU incorporation during the first half of S-phase across two copies of chr. X in human female cell line GM12878 (data from ENCODE consortium [57]).

Despite differences in replication timing between active *spretus* and inactive *musculus* X chromosomes, the relative order in which different regions within each chromosome complete replication is preserved, with earlier replicating regions corresponding to regions of higher MCM density (Fig 9A, right). This suggests that, although the chromosome-wide replication timing on the active and inactive X chromosomes differs, replication timing *within* each chromosome is still determined by variation in MCM density along that chromosome. Because this orderly replication pattern for the mouse Xi contrasts with the random initiation pattern that has been reported for the human Xi [57], we re-examined Xi replication kinetics of in the human female lymphoblastoid cell line GM12878. This cell line is well-suited for this purpose because (1) high-quality haplotype-resolved genome sequences are available [58], (2) the active

X chromosome is derived almost exclusively from one of the two parents [59]; and (3) high resolution replication profiles of these cells have been obtained by repli-seq [60]. In the repli-seq study, BrdU incorporation was measured at several time points during S-phase, assuring similar resolution of replication kinetics within early and late S-phase. Our analysis of the repli-seq dataset showed that, although the inactive human X replicates later than the active X, the relative order of replication along each chromosome is comparable (Fig 9B), exactly as we saw in mouse. Overall, our observations suggest a model in which chromosome-wide replication delay on the inactive X is regulated by control of origin firing, but replication within each X chromosome is regulated by origin licensing. Furthermore, given that transcription is largely abolished on the Xi, this suggests that the spatial pattern of both MCM loading and replication order on the Xi are independent of transcription. Finally, this result demonstrates that heterochromatin is not an impediment for MCM loading, but rather for activation, of licensed origins.

## Discussion

Loading of MCM helicase complexes across the genome during G1 is the central event in origin licensing, i.e. in specifying which sites may serve to initiate replication in the subsequent S phase. By fusing the MCM2 subunit of the complex to micrococcal nuclease, we were able to identify these sites at nucleotide resolution, quantify their relative levels of licensing in the population, and examine the anatomy of individual sites. By applying the same technique in three different organisms in which direct measurements of replication were publicly available, we were able to confirm our results, determine their implications for replication activity, and use them to elucidate differences in origin licensing strategies across species.

One question that has attracted significant attention over the last two decades is whether multiple MCM complexes are loaded at individual origins in a single cell and, if so, whether increased numbers of helicases are associated with higher origin activity [45, 61]. For example, the observation that up to ten times more DNA is immunoprecipitated in MCM- as compared to ORC-ChIP studies, supported the notion that multiple MCM complexes were loaded at individual origins [62]. Budding yeast is an ideal organism in which to address this question, due to sequence specificity of its replication origins; however, Mcm-ChIP, while adequate for determining the relative MCM abundance among the origins, lacks the necessary resolution to unambiguously resolve the exact number of loaded MCM complexes. The high resolution of the ChEC technique overcomes this barrier and thus enabled us to address the question directly, as follows: First, we found that approximately half (167/343) of origins listed in SGD contained a single MCM DH, as defined by the criteria described in the Results. Second, our Southern blot and qPCR analysis of an MCM-ChEC digestion time course at the highly efficient ARS1103, which features a single MCM DH, demonstrate that an MCM DH has been loaded at this origin in at least 90% of the cells. Third, both the median and average MCM abundance is higher at single than at multiple-footprint origins. And fourth, high MCM abundance origins are skewed toward single MCM-footprint origins. These results demonstrate that these single footprint origins contain one or zero MCM DH per origin, and that differences in MCM abundance at these origins reflect the fraction of cells in which a single DH has been loaded rather than the number of complexes that have been loaded per origin, as previously proposed [45]. Do the origins that contain multiple MCM footprints have multiple MCM DHs per origin? While it is certainly possible that some cells contain multiple MCM complexes at a single origin, we favor the idea that multiple footprints at single origins generally arise as a composite signal from multiple cells, each of which contains a single MCM DH. First, origins that have multiple footprints have, on average, *less* total MCM signal per origin

than origins that have just one footprint; thus, it is counterintuitive that licensing of the former class of origins should involve loading of *more* MCM complexes per origin than are loaded for the latter class. Second, because an origin like ARS1103 can be among the most active in the genome while depending on a single MCM complex for that activity, a single MCM complex is clearly sufficient for origin function. And third, there is a paucity of specific DNA fragments that are predicted to be generated if multiple MCM complexes are present at a single origin in a single cell (S3 Fig). Thus, we favor the idea that most of the 343 budding yeast origins have either one or zero loaded MCMs.

What are the implications of our findings in budding yeast on the pattern on MCM loading in fission yeast and mammals? In contrast to the single footprints at many highly active origins in *S. cerevisiae*, which allowed us to conclude that either one or zero MCM DHs are loaded at these origins in any individual cell, we typically saw 4–16 contiguous DH footprints distributed along 0.6–1.6 kb of DNA at fission yeast origins, making the question how many MCM complexes are present in a single cell ambiguous. For simplicity, we favor the notion that in *S. pombe*, just as in *S. cerevisiae*, most licensed origins contain a single MCM complex, and that the multiple footprints at *S. pombe* origins reflect a composite of single footprints from different cells. For the same reason, we suggest that the increased MCM density along the 100 kb initiation zone in mammals (see Results and [49]) reflect the contributions of footprints originating from different cells.

Our data strongly indicate that uneven distribution of MCM along the chromosome underlies the basic chromosomal replication pattern in both yeast and mammals, because the pattern of MCM distribution is broadly recapitulated in the replication program. In budding yeast, MCM abundance as assessed by MCM-ChIP has been reported to correlate with the metrics that incorporate both the inter-origin distance and replication timing, though the level of correlation was rather modest (r = 0.42 genome wide) [45]. In our work the relationship between MCM abundance and replication pattern in budding yeast is clearly seen in replication simulations, which assign the probability of replication initiation at specific chromosomal sites according to MCM distribution along the chromosome, capturing both the MCM abundance at individual origins and their distribution with relation to each other along the chromosome. While simulations recapitulate the replication program, if we ignore inter-origin distance and their distribution along the chromosome, and plot MCM abundance against the replication timing for each origin using the same dataset, we found no correlation between the two (S18A Fig). This is likely due to the fact that the level of replication initiation activity at an origin depends both on active replication at that origin and passive replication from the nearby origins, which is captured in simulations but not when directly relating MCM abundance with replication timing across the origin. Another way to evaluate the relationship between MCM abundance and active replication at individual origins is to measure replication in HU, which slows down fork movement and thus measures active replication at an origin while minimizing passive replication from the adjacent origins. Indeed, we found that replication initiation at ORI DB origins (excluding repetitive origins), as judged by accumulation of ssDNA in HU [46], correlates with both MCM2-ChEC (R = 0.65) and published MCM2-Chip datasets (R = 0.57 and 0.56 [44,45] for Das et al and Belsky et al, respectively), as shown in S18B Fig (see S2 Table for the dataset used for these calculations).

Our data also emphasize that MCM loading cannot be the sole determinant of replication timing, as we clearly identify regions that replicate later than would be expected based on their MCM abundance; most notable among these regions is the inactive X-chromosome in mammals. We do not suggest that the inactive X-chromosome is the only site in mammalian genome subject to this sort of post-licensing regulation, but it constitutes a particularly clear example: The equal loading of MCM and the identical sequence of the two chromosomes

made it simple to attribute delayed replication of the Xi to a post-licensing step, whereas similar, heterochromatic regions on autosomes that replicate later than expected based on their MCM abundance are more difficult to identify. Another potential form of post-licensing regulation, the magnitude of which is difficult to quantify, may further enforce the replication pattern laid out by uneven chromosomal MCM distribution: If, after activating one licensed origin, factors required for origin firing tend to diffuse to nearby origins, then the process of licensing origins more heavily in one region of the genome will simultaneously promote firing of those origins. Indeed, this possibility blurs the distinction between regulation of origin activity at the levels of licensing and firing. Despite these limitations, given that replication timing along the chromosomes broadly reflects chromosomal MCM distribution in all four species, our results provide strong support for a model of stochastic firing of licensed origins, as has been proposed by others [31–40], as the primary driver of chromosomal replication timing.

## Materials and methods

### Cell culture and media

All *S. cerevisiae* strains were in S288C background. *S. cerevisiae* experiments were carried out using standard YPD (yeast peptone dextrose) medium [2% (wt/vol) glucose, 1% yeast extract, 2% (wt/vol) peptone]. *S. pombe* experiments were carried out in Yeast Extract Supplemented 3% Dextrose (YES) [63]; logarithmically growing cells were arrested in early S-phase by adding 15 mM hydroxyurea and incubating for 3 hours. Strains are listed in S3 Table.

### Tagging Mcm with MNase

Human and mouse MCM2 genes, tagged with micrococcal nuclease at the C-terminus using the same tag that was employed in yeast were cloned into a lentiviral vector. The plx302-mMCM2MNase was made as follows: Mouse MCM2 cDNA was amplified from plasmid mEmeraldMCM2-N-22, which we received as a gift from Michael Davidson (Addgene plasmid # 54164; http://n2t.net/addgene:54164; RRID:Addgene_54164). 3X FLAG-MNase was amplified from pGZ108 (pGZ108 (pFA6a-3FLAG-MNase-kanMX6), which we received as a gift from Steven Henikoff (Addgene plasmid # 70231; http://n2t.net/addgene:70231; RRID: Addgene_70231).

The lentiviral plasmid plx302 was digested with NdeI/NheI and 530bp of plx302 starting from the NdeI cut site was also PCR amplified. All the fragments were assembled by Gibson Assembly as per the manufacturer's instructions (NEB # E2611S).

The plx302-hMCM2-Mnase was made in two stages: First, a Gateway destination vector containing FL-MNase immediately downstream of the attR2 recombination site was constructed.

For this, the Gateway destination vector plx302 was linearized with NdeI/NheI, 3XFLAG-MNase was amplified from plx302-mMCM2-MNase and cmR-ccdB was amplified from plx302. These fragments were assembled using Gibson Assembly as before to create plx302-FL-MNase Gateway destination vector. Next, human MCM2 cDNA from plx304-hMCM2-V5 (a gift from Dr. Patrick Paddison, FHCRC) was shuttled to the Gateway donor vector pDONR221 (a gift from Dr. Valeri Vasioukhin, FHCRC) using BP clonase (ThermoFisher Scientific #11789100) as per the manufacturer's instructions. The resulting entry clone was then used to shuttle hMCM2 cDNA to the plx302-FL-MNase by LR clonase (ThermoFisher Scientific # 1253810) to obtain the final expression vector. All the final vectors were sequenced.

### Primers used for mutagenesis of mouse MCM2

1. XXI-57: GAGCATGGCAGGGCCCAGGCTGGAGATCC

2. XXI-58: ACCCACTCGCGCACCGAGTGGCCCTTGAGG

### Primers used to amplify DNA segments for Gibson cloning of mouse MCM2

1. XXI-51: CTTGGCAGTACATCAAGTGTATCATATGCCAAGTACGCCCCCTATTG

2. XXI-52:
   GAGAGACTCAGAAGACTCCGCCATCATAGTGACTGGATATGTTGTGTTTTAC

These two were used to PCR out CMV promoter

3. XXI-53:
   GTAAAACACAACATATCCAGTCACTATGATGGCGGAGTCTTCTGAGTCTCTC

4. XXI-54: TTAACCCGGGGATCCGTCGACCGAACTGCTGTAGGATCAGTTTGC

   These two were used to amplify mouse MCM2 cDNA

5. XXI-55: GCAAACTGATCCTACAGCAGTTCGGTCGACGGATCCCCGGG

6. XXI-56: CTTAACGCGCCACCGGTTAGCGCTACTGGCCGCTATCGGCGTTATC

   These two were used to amplify FL-MNase

### List of sequencing primers used to confirm sequences of mouse MCM2

1. XXI-59: CATTGACGTCAATAATGACGTATGTTCC

2. XXI-60: TGGCTAACTGTCGGGATCAACAAG

3. XXI-61: TGATGAAGATGTGGAGGAGCTGAC

4. XXI-62: GTTGCTGCAGATCTTTGACGAGGC

5. XXI-63: AGCTGACCGGCATTTACCATAATAAC

6. XXI-64: GTGTCTCATTGACGAGTTTGACAAG

7. XXI-65: CTCAACCAGATGGACCAGGATAAAG

8. XXI-66: CCTGAGAAGGATCTGATGGACAAG

9. XXI-67: GTTCGATAAAGGCCAACGCAC

10. XXI-68: GTTGCGTCAGCAAACACAGTG

### Primers used to amplify DNA segments for Gibson cloning of plx302-FL-MNase Gateway destination vector

1. Plx302_FLMNase_frag1_fwd:

TGCCCACTTGGCAGTACATCAAGTGTATCATATGCCAAGTACGCCCCCTATTG

2. Plx302_FLMNase_Frag1 rev:

ATCTTTAATTAACCCGGGGATCCGTCGACCAACCACTTTGTACAAG

These two were used to PCR CMV-ccdb fragment from plx302.

3. Plx302_FLMNase Frag2 fwd:

CGTTTCTCGTTCAGCTTTCTTGTACAAAGTGGTTGGTCGACGGATCCCCGGGT
TAATTAAAGATTACAAG

4. Plx302MNase_Frag2 rev:

TTGTCGACTTAACGCGCCACCGGTTAGCGCTAGCTCATTACTACTGGCCGCTA
TCGGCG

These two were used to PCR out FL-MNase from plx302-mMCM2-FLMNase **Sequencing
primers used for humanMCM2 plasmid**

1. XXI-60: TGGCTAACTGTCGGGATCAACAAG

2. XXI-67: GTTCGATAAAGGCCAACGCAC

3. XXI-68: GTTGCGTCAGCAAACACAGTG

## Human and mouse MCM2-FLAG-MNase cell lines

Cells were transduced with lentiviral supernatant containing plx302-MCM2-FL-MNase,
infected cells were selected out on Puromycin (2.5μg/ml) and expanded for further experi-
ments. The expression of tagged MCM2 in HeLa and Patski cells was confirmed by Western
blot using antiMCM2 rabbit polyclonal antibody (HPA031495 Sigma) at 1:1,000 dilution.

## Yeast ChEC protocol

In budding yeast, ChEC-seq was carried out as previously described [43]. Briefly,

HU arrest in budding yeast was carried out by adding 200 mM hydroxyurea to logarithmi-
cally growing cultures for 50 minutes. Fission yeast with MNases-tagged MCM2 were trans-
ferred from log phase to 15 mM HU for 3 hours and then processed the same as *S. cerevisiae*.
Two biological replicas were prepared for each genotype.

Cells were centrifuged at 1,500 x g for 2 mins, and washed twice in cold Buffer A (15 mM Tris
pH 7.5, 80 mM KCl, 0.1 mM EGTA) without additives. Washed cells were carefully resuspended
in 570 μL Buffer A with additives (0.2 mM spermidine, 0.5 mM spermine, 1 mM PMSF, ½ cOm-
plete ULTRA protease inhibitors tablet, Roche, per 5 mL Buffer A) and permeabilized with 0.1%
digitonin in 30°C water bath for 5 min. Permeabilized cells were cooled at room temperature for 1
min and 1/5th of cells were transferred in a tube with freshly made 2x stop buffer (400 mM NaCl,
20 mM EDTA, 4 mM EGTA)/1% SDS solution for undigested control. Micrococcal nuclease was
activated with 5.5 μL of 200 mM CaCl2 at various times (30 sec, 1 min, 5 mins, and 10 mins) and
the reaction stopped with 2x stop buffer/1% SDS. Once all time points were collected, proteinase K
was added to each collected time points and incubated at 55°C water bath for 30 mins. DNA was
extracted using phenol/chloroform and precipitated with ethanol. Micrococcal nuclease digestion
was analyzed via gel electrophoresis prior to proceeding to library preparation. Library was pre-
pared as previously described used using total DNA, without any fragment size selection [43].

## Mammalian ChEC protocol

Cells were grown in 6 well plates in Dulbeco Modified Eagle Medium (DMEM) supplemented
with 10% Fetal Calf Serum until 100% confluent, trypsinized and washed with a Wash Buffer
(20mM HEPES, 110mM Potassium Acetate, 5mM Sodium Acetate, pH to 7.3 with NaOH). Cells

were permeabilized with 0.02% digitonin in the Wash Buffer along with protease inhibitors (Roche # 04693159001) for 5 mins at RT. $CaCl_2$ was added to a final concentration of 2.5mM for differing amounts of time, as indicated. The MNase activity was stopped by adding equal volume of a 2X Stop Buffer (400mM NaCl, 20mM EDTA, 4mM EGTA, 2% SDS). Proteinase K was added to a final concentration of 0.4mg/ml and samples incubated for 1 hr at 50˚ C. DNA was isolated using standard Phenol:Chloroform extraction and air dried DNA pellets resuspended in 0.1X TE (pH 8). All samples were treated with RNase A (0.3 µg/ml) before being processed for library preparation. Two biological replicas were prepared for both human and mouse cells.

### Exogenouse MNase-seq

We carried out MNase-Seq in both S. cerevisiae and in S. pombe using the same method we have previously described for S. cerevisiae [43]. Cells grown to log phase in rich medium, Yeast Peptone Agar with 2% glucose (YEPD) for S. Cerevisiae or Yeast Extract with Supplements with 3% glucose (YES), from an overnight 25 mL culture were synchronized with 3 µM alpha-factor for 1.5 hrs. at 30˚ C (S. cerevisiae) or with 15 mM hydroxyurea for 2 hours in S. pombe, which are the same conditions we used to prepare cells for ChEC. Arrested cells were crosslinked with 1% formaldehyde for 30 min at room temperature water bath with shaking. Formaldehyde was quenched with 125 mM glycine and cells were centrifuged at 3000 rpm for 5 min. Cells were washed twice with water and resuspended in 1.5 mL Buffer Z (1 M sorbitol, 50 mM Tris-HCl pH 7.4) with 1 mM beta-mercaptoethanol (1.1 µL of 14.3 M beta-mercaptoethanol diluted 1:10 in Buffer Z) per 25 mL culture. Cells were treated with 100 µL 20 mg/mL zymolyase at 30˚ C for 20–30 min depending on cell density. Spheroplasts were centrifuged at 5000 rpm for 10 min and resuspended in 5 mL NP buffer (1 M sorbitol, 50 mM NaCl, 10 mM Tris pH 7.4, 5 mM MgCl2, 1 mM CaCl2) supplemented with 500 µM spermidine, 1 mM beta-mercaptoethanol and 0.075% NP-40. Nuclei were aliquoted in tubes with varying concentrations of micrococcal nuclease (Worthington), mixed via tube inversion, and incubated at room temperature for 20 mins. Chromatin digested with 1.9 U– 7.5 U micrococcal nuclease per 1/5th of spheroplasts from a 25 mL culture yielded appropriate mono-, di-, trinucleosome protected fragments for next-generation sequencing. Digestion was stopped with freshly made 5x stop buffer (5% SDS, 50 mM EDTA) and proteinase K was added (0.2 mg/ml final concentration) for an overnight incubation at 65˚ C to reverse crosslinking. DNA was extracted with phenol/chloroform and precipitated with ethanol. Micrococcal nuclease digestion was analyzed via gel electrophoresis prior to proceeding to library preparation. Sequencing libraries for both MNase-seq and ChEC-seq were prepared as described [21].

### Southern blot for ARS1103

Cells carrying Mcm2-MNase were arrested in alpha factor, spun down, permeabilized with digitonin and exposed to calcium for 0, 15 s, 30 s, 1 min, 5 min and 10 min. DNA was extracted, digested with Xmn1, separated on a 1.75% agarose gels, and analyzed by standard Southern blotting technique [64].

### pPCR for time course of MNase digestion for ARS1103

Cells carrying Mcm2-,4- and 6-MNase were subjected to ChEC. DNA was extracted and analyzed for cleavage at ARS1103 by qPCR using the following primers, as depicted in Fig 3:
    qPCR_Left Nucleosome_5p_2: CACTTAACTTGTTATAATTCTCCC
    qPCR_Right Nucleosome_3p_1: CCATTCTGGTAGTTTTAATGTATTG
    5p_Tor2_qPCR_2: GCACAAAGACCACAAAGTCG

3p_Tor2_qPCR_2: GGGGATAGAGAACTAACAAAAGCA pair of primers that amplify ~ 100 bp located ~ 3 kb 3' of the ARS1103 was used to normalize the ARS1103 signal with the input DNA.

## Analysis of the hybrid *M.musculus/M.spretus* Patski cell line

A hybrid musculus-spretus genome was generated in silico by (1) identifying all sites in which the musculus (mm10) and spretus (SPRET_EiJ) genomes differed by a single nucleotide substitution and (2) replacing all such sites in the mm10 genome with the corresponding SPRET_EiJ sequence. fastq files were then aligned to the mm10 genome and the hybrid mm10—SPRET_EiJ genome using the bwa alignment software (version 0.7.17) [65] and processed with Picard's tools as described above. Reads were retained only if they (1) contained at least one of the singlenucleotide differences that were used to generate the hybrid mm10—SPRET_EiJ genome and (2) had no mismatches with the genome to which they were aligned. Reads that did not match these criteria were filtered out using Picard's FilterSamReads tool (version 2.21.6) (http://broadinstitute.github.io/picard/). Per-base pair read depths were then determined with BedTools' genomecov tool, Version 2.29.1 [66], with those coming from the filtered bam files generated from the mm10 and hybrid genome alignments constituting the read depths assigned to the *musculus* and *spretus* genomes, respectively.

## *M. spretus* and *M. musculus* chromosome DNA replication and MCM-ChEC analysis

Replication was measured by S-seq [67]. Patski cells were stained with propidium iodine and sorted according to DNA content into G1 and S-cells using flow cytometry. DNA was extracted from each cell fraction, fragmented by sonication and sequenced. Musculus and spretus chromosome replication was expressed as a ratio of S to G1 read depths for reads that mapped to each of the two genomes. MCM abundance at musculus and spretus chromosomes was calculated as a ratio of Mcm-ChEC to sonicated G1 DNA read depths for reads that mapped to each of the two genomes. Two biological replicas were analyzed.

## Sequencing

Sequencing was performed using an Illumina HiSeq 2500 in Rapid mode employing a pairedend, 50 base read length (PE50) sequencing strategy. Image analysis and base calling was performed using Illumina's Real Time Analysis v1.18 software, followed by 'demultiplexing' of indexed reads and generation of FASTQ files, using Illumina's bcl2fastq Conversion Software v1.8.4.

## Sequence analysis

fastq files were aligned to genome assemblies for *Saccharomyces cerevisiae* (sacCer3), *Schizosaccharomyces pombe* (ASM294v2), *Homo sapiens* (GRCh38/hg38), or *Mus musculus* (mm10) using bwa alignment software (version 0.7.17) [65] with the -n 1 option, which causes reads that map to more than one location to be randomly assigned to one of those locations. bam files were then processed with Picard's CleanSam, SortSam, FixMateInformation, AddOrReplaceReadGroups and ValidateSamFile tools (version 2.21.6; http://broadinstitute.github.io/picard/). Per-base pair read depths were then determined with BedTools' genomecov tool, Version 2.29.1, using the -d and -split options [66]. Information regarding numbers of reads mapped for each sample is in S4 Table. An example of taking a single pair of fastq files all the

way to generation of a heat map showing MCM2-ChEC footprints is provided in the S1 Source File for Fig 1.

## Replication simulations

The replication simulations had 3 parameters: (1) the bin size into which MCM2-ChEC will be grouped to generate probability distributions; (2) the number of MCM complexes that are activated at the beginning of S phase; and (3) the distance replication forks are allowed to go before another batch of MCM complexes, which will correspond to the number of converged forks, will be activated. For these parameters, we chose 1 kb, 300, and 5 kb, respectively. The second parameter can be thought of as reflecting the number of MCM complexes in the cell, and the third parameter can be thought of as reflecting the frequency with which replication factors are "recycled" after diffusion. Replication simulation results were very robust to variation in these parameters, including completely eliminating "recycling".

Per-base pair read depths from *cerevisiae* MCM2-ChEC data sets were binned in 1 kb bins across the genome and these numbers were used to generate probability distributions. 300 MCM complexes were chosen from this probability distribution, the central base pair from each 1 kb bin was assigned the number 0, base pairs to the right of this base pair were assigned the numbers +1, +2, +3, etc., and base pairs to the left were assigned corresponding negative numbers. These numbers increase in absolute value until they either (1) run off the end of the chromosome or (2) converge with numbers coming from another activated MCM complex. At this point, every base pair in the genome will have been assigned a number, which corresponds to the order in which it has been "replicated", and we then count the number of replication forks that would have converged and activate that number of MCM complexes, chosen at random from the original probability distribution after excluding (1) MCM complexes that have already been activated and (2) MCM complexes that have been replicated. This process is repeated until the entire genome has been replicated by a replication fork that has not traveled more than 5 kb. 1000 "cells" were simulated and from these, the replication pattern of the corresponding population was inferred.

## Determination of replication initiation and termination zones in *S. pombe*

Replication initiation zones were defined for *S. pombe*, on the basis of publicly available smoothed polymerase usage wig.bed files (NCBI Gene Expression Omnibus accession number GSE62108), as transitions from majority polδ usage to majority polε usage in the synthesis of the new Watson strand, with the opposite transitions constituting termination zones. To avoid obfuscation of signal due to close proximity of zones of opposite types, initiation and termination zones were included in the analysis only if they were at least 5 kb from the nearest termination and initiation zones, respectively.

## GM12878 repli-seq replication and histone modifications

We downloaded fastq files from the ENCODE portal [60] (https://www.encodeproject.org/) with the following identifiers: ENCFF001GNU, ENCFF001GNY, ENCFF001GOB, ENCFF001GOG, ENCFF001GOM, ENCFF001GOO, ENCFF000BBP, ENCFF000BBQ, ENCFF000BBT, ENCFF000BBW, ENCFF000BBY, ENCFF000BBZ, ENCFF000BCE, ENCFF000BCI, ENCFF000BCT, ENCFF000BCU, ENCFF000BCZ, ENCFF000BDA, ENCFF000BDH, ENCFF000BDN, ENCFF001FKI, ENCFF001FKM, ENCFF001FLD, ENCFF001FLG, ENCFF248WCN, ENCFF377VCZ, ENCFF485LYA, ENCFF588FTN, ENCFF625VRT. These were then aligned and processed as described above.

## Supporting information

**S1 Fig. Streaking out strains on rich medium (YEPD) indicates that tagging MCM2, MCM4 and MCM6 with MNase did not affect growth rates.** Flow cytometry profiles show log phase cultures stained for DNA content, with the two major peaks representing cells in G1 and G2.
(TIF)

**S2 Fig. Reproducibility of *S. cerevisiae* ChEC when tagging different subunits of the MCM complex.** Genome-wide signal, in 100 base pair bins, plotted with Mcm2-ChEC on the x axis in all three cases, and on the y axis either Mcm4-ChEC (r = 0.96), Mcm6-ChEC (r = 0.94) or free MNase (r = 0.09). All fragment sizes were included in the calculations.
(TIF)

**S3 Fig. Fragment sizes in libraries from *S. cerevisiae* strains with tagged Mcm2, Mcm4 and Mcm6 are comparable and consistent with the 62 bp of DNA shown to be protected by the MCM helicase complex by cryo-EM.** Fragment sizes for the peaks of Mcm2-, Mcm4- and Mcm6-tagged strains are 58, 50 and 62 base pairs, respectively, and median fragment lengths for fragments < = 100 base pairs were 61, 58 and 70 base pairs, respectively.
(TIF)

**S4 Fig. Five panels with 343 origins from Saccharomyces Genome Database (SGD) presented as heat maps and the corresponding MCM abundance distribution for each origin using reads in the 50–100 bp size range.** The markings on the plots are the same as for the 12 origins on chromosome IV in Fig 3. 9 origins listed in SGD were omitted from this analysis due bioinformatic difficulties, such as ambiguous mapping locations. 187 of the remaining 343 origins contained ACS sequences, and these are indicated with dotted lines. Each figure shows a 3 kb span on the X axis. Read depths are generally capped by setting the lightest color (highest read depth) to a per-base pair read depth that is less than the maximum read depth. This is done to enhance lower intensity signals. Capped read depths for each origin are listed here with maximum (uncapped) read depths following in parentheses: ARS1002: 160 (1285); ARS1003: 17 (143); ARS1004: 121 (975); ARS1005: 144 (1152); ARS1006: 12 (97); ARS1007: 53 (430); ARS1008: 69 (559); ARS1009: 56 (449); ARS1010: 26 (210); ARS1011: 128 (1025); ARS1012: 161 (1293); ARS1013: 161 (1293); ARS1014: 48 (384); ARS1015: 78 (631); ARS1016: 3 (27); ARS1017: 9 (77); ARS1018: 64 (513); ARS1019: 63 (510); ARS1020: 74 (593); ARS1021: 29 (234); ARS1022: 38 (304); ARS1023: 406 (3250); ARS1024: 21 (169); ARS103: 23 (190); ARS104: 41 (332); ARS105: 19 (159); ARS106: 14 (113); ARS107: 85 (683); ARS108: 16 (135); ARS109: 100 (804); ARS110: 15 (121); ARS1102: 7 (7); ARS1103: 397 (3177); ARS1104: 1 (14); ARS1106: 75 (603); ARS1109: 55 (443); ARS111: 658 (5270); ARS1112: 130 (1044); ARS1113: 94 (752); ARS1114: 39 (315); ARS1115: 80 (647); ARS1116: 24 (199); ARS1118: 52 (416); ARS112: 34 (275); ARS1120: 107 (858); ARS1123: 111 (892); ARS1125: 67 (541); ARS1126: 7 (56); ARS1127: 67 (542); ARS1200-1: 6655 (53244); ARS1200-2: 5457 (43658); ARS1202: 173 (1390); ARS1206: 65 (522); ARS1207: 104 (832); ARS1208: 6 (50); ARS1209: 106 (854); ARS1210: 72 (578); ARS1211: 13 (111); ARS1212: 81 (653); ARS1212.5: 12 (98); ARS1213: 40 (326); ARS1214: 1 (8); ARS1215: 21 (169); ARS1216: 162 (1302); ARS1217: 146 (1175); ARS1218: 131 (1055); ARS1219: 11 (95); ARS1220: 219 (1758); ARS1221: 5 (5); ARS1223: 52 (416); ARS1226: 26 (212); ARS1228: 1 (11); ARS1230: 1 (10); ARS1232: 64 (517); ARS1233: 7 (58); ARS1234: 42 (343); ARS1235: 86 (693); ARS1238: 25 (201); ARS1303: 16 (130); ARS1304: 17 (139); ARS1305: 7 (63); ARS1307: 59 (472); ARS1307.5: 14 (116); ARS1308: 8 (66); ARS1309: 235 (1884); ARS1310: 70 (561); ARS1312: 33 (269); ARS1316: 14 (116); ARS1317: 5 (5); ARS1319: 36 (290); ARS1320: 128 (1024); ARS1321: 1 (12); ARS1322: 38 (304); ARS1323:

32 (260); ARS1324: 32 (258); ARS1325: 15 (120); ARS1327: 58 (464); ARS1328: 104 (833); ARS1329: 9 (78); ARS1330: 28 (226); ARS1331: 9 (76); ARS1332: 180 (1443); ARS1333: 15 (122); ARS1405: 81 (648); ARS1406: 198 (1589); ARS1407: 167 (1342); ARS1409: 1 (8); ARS1410: 24 (193); ARS1411: 247 (1976); ARS1412: 74 (598); ARS1413: 44 (353); ARS1414: 47 (380); ARS1415: 36 (288); ARS1416: 1 (8); ARS1417: 56 (455); ARS1419: 52 (418); ARS1420: 97 (782); ARS1421: 11 (95); ARS1422: 13 (107); ARS1423: 1 (8); ARS1424: 189 (1516); ARS1425: 7 (7); ARS1426: 9 (77); ARS1427: 75 (602); ARS1501: 26 (212); ARS1502: 35 (280); ARS1507: 27 (222); ARS1508: 23 (190); ARS1509: 24 (195); ARS1510: 28 (224); ARS1511: 115 (926); ARS1512: 125 (1000); ARS1513: 29 (236); ARS1516: 71 (569); ARS1518: 55 (442); ARS1519: 118 (945); ARS1521: 59 (476); ARS1523: 24 (197); ARS1524: 24 (199); ARS1526: 75 (601); ARS1528: 38 (310); ARS1529: 72 (581); ARS1531: 152 (1218); ARS1603: 5 (42); ARS1604: 20 (162); ARS1605: 81 (653); ARS1607: 28 (230); ARS1608: 228 (1825); ARS1611: 4 (4); ARS1612: 43 (348); ARS1614: 51 (413); ARS1617: 36 (288); ARS1618: 68 (549); ARS1619: 44 (355); ARS1620: 4 (4); ARS1621: 50 (404); ARS1622: 107 (856); ARS1623: 126 (1010); ARS1624: 7 (61); ARS1625: 109 (873); ARS1626: 36 (289); ARS1627: 34 (277); ARS1628: 32 (262); ARS1629: 1 (10); ARS1630: 48 (390); ARS1631: 28 (225); ARS1632: 76 (614); ARS1633: 70 (561); ARS1634: 107 (856); ARS1635: 55 (445); ARS201: 168 (1345); ARS202: 42 (342); ARS203: 43 (349); ARS206: 227 (1818); ARS207: 50 (404); ARS208: 73 (590); ARS209: 4 (33); ARS211: 66 (531); ARS212: 4 (37); ARS213: 139 (1119); ARS214: 36 (288); ARS215: 56 (451); ARS216: 253 (2029); ARS217: 12 (100); ARS220: 19 (155); ARS221: 6 (52); ARS222: 96 (769); ARS224: 194 (1556); ARS225: 129 (1034); ARS228: 78 (626); ARS230: 192 (1537); ARS231: 11 (92); ARS301: 4 (4); ARS302: 5 (42); ARS303: 5 (42); ARS304: 11 (90); ARS305: 4 (39); ARS306: 424 (3399); ARS307: 126 (1010); ARS308: 44 (359); ARS309: 85 (682); ARS310: 122 (983); ARS313: 33 (264); ARS314: 42 (336); ARS315: 208 (1669); ARS316: 23 (191); ARS317: 21 (169); ARS318: 32 (262); ARS320: 5 (42); ARS403: 215 (1723); ARS404: 68 (546); ARS405: 48 (384); ARS406: 24 (195); ARS409: 155 (1243); ARS409.5: 2 (21); ARS410: 158 (1264); ARS411: 1 (12); ARS412: 63 (510); ARS413: 86 (691); ARS414: 14 (112); ARS415: 32 (260); ARS416: 203 (1631); ARS417: 212 (1697); ARS418: 38 (308); ARS419: 199 (1599); ARS420: 60 (481); ARS421: 13 (104); ARS422: 52 (423); ARS423: 19 (156); ARS425: 69 (552); ARS427: 7 (58); ARS428: 46 (371); ARS429: 14 (119); ARS430: 285 (2284); ARS431: 35 (286); ARS432: 229 (1839); ARS433: 176 (1414); ARS434: 19 (156); ARS435: 114 (915); ARS436: 7 (7); ARS439: 17 (143); ARS440: 15 (121); ARS442: 6 (6); ARS443: 43 (350); ARS446: 48 (390); ARS450: 157 (1262); ARS451: 20 (165); ARS452: 125 (1001); ARS453: 147 (1182); ARS502: 2 (17); ARS503: 26 (210); ARS504: 7 (7); ARS507: 144 (1154); ARS508: 177 (1419); ARS510: 85 (681); ARS511: 48 (388); ARS512: 22 (182); ARS513: 1 (8); ARS513.5: 6 (49); ARS513.7: 12 (100); ARS514: 96 (770); ARS515: 48 (391); ARS516: 18 (146); ARS517: 183 (1470); ARS518: 93 (747); ARS519: 1 (8); ARS520: 74 (599); ARS521: 3 (25); ARS522: 36 (290); ARS523: 81 (651); ARS600: 85 (686); ARS600.4: 106 (849); ARS601: 118 (950); ARS602: 118 (950); ARS603: 121 (972); ARS603.1: 3 (24); ARS603.5: 34 (278); ARS604: 1 (15); ARS605: 178 (1428); ARS606: 36 (292); ARS607: 73 (591); ARS608: 18 (147); ARS609: 88 (711); ARS701: 1 (9); ARS702: 39 (315); ARS704: 24 (196); ARS706: 4 (38); ARS707: 15 (121); ARS709: 4 (32); ARS710: 24 (198); ARS712: 104 (832); ARS714: 59 (479); ARS715: 4 (38); ARS716: 292 (2338); ARS717: 10 (82); ARS718: 74 (595); ARS719: 52 (418); ARS720: 26 (208); ARS721: 883 (7067); ARS722: 34 (279); ARS723: 34 (279); ARS724: 1 (13); ARS727: 131 (1051); ARS728: 17 (141); ARS729: 33 (264); ARS731: 98 (791); ARS733: 15 (127); ARS734: 62 (502); ARS735: 1 (14); ARS736: 98 (788); ARS737: 107 (861); ARS801: 55 (446); ARS802: 35 (283); ARS805: 12 (97); ARS806: 42 (341); ARS807: 142 (1141); ARS808: 6 (6); ARS809: 57 (460); ARS810: 504 (4033); ARS811: 313 (2505); ARS813: 43 (345); ARS814: 5 (5); ARS815: 143 (1145); ARS818: 90 (722); ARS820: 32 (259); ARS822: 75 (603); ARS824: 120 (966); ARS902: 88 (711); ARS904: 151

(1210); ARS907: 60 (485); ARS909: 171 (1374); ARS910: 6 (6); ARS911: 11 (93); ARS912: 32 (262); ARS913: 141 (1129); ARS914: 50 (405); ARS915: 50 (405); ARS918: 1 (8); ARS919: 26 (213); ARS920: 23 (187); ARS921: 6 (6); ARS922: 231 (1853); ARS923: 30 (247); ARS1002: 6 (49); ARS1003: 1 (13); ARS1004: 9 (72); ARS1005: 15 (120); ARS1006: 3 (28); ARS1007: 5 (41); ARS1008: 3 (31); ARS1009: 3 (28); ARS1010: 2 (18); ARS1011: 9 (77); ARS1012: 8 (69); ARS1013: 8 (69); ARS1014: 3 (27); ARS1015: 6 (54); ARS1016: 4 (4); ARS1017: 1 (10); ARS1018: 5 (40); ARS1019: 7 (59); ARS1020: 3 (24); ARS1021: 1 (10); ARS1022: 3 (26); ARS1023: 4 (33); ARS1024: 2 (22); ARS103: 7 (7); ARS104: 2 (22); ARS105: 4 (33); ARS106: 1 (13); ARS107: 3 (30); ARS108: 2 (22); ARS109: 10 (85); ARS110: 3 (24); ARS1102: 2 (2); ARS1103: 35 (285); ARS1104: 2 (2); ARS1106: 4 (39); ARS1109: 8 (68); ARS111: 25 (200); ARS1112: 8 (71); ARS1113: 3 (29); ARS1114: 6 (54); ARS1115: 2 (20); ARS1116: 2 (21); ARS1118: 2 (22); ARS112: 6 (6); ARS1120: 1 (11); ARS1123: 25 (201); ARS1125: 8 (69); ARS1126: 7 (7); ARS1127: 4 (38); ARS1200-1: 603 (4830); ARS1200-2: 747 (5978); ARS1202: 8 (69); ARS1206: 6 (52); ARS1207: 8 (69); ARS1208: 5 (5); ARS1209: 5 (42); ARS1210: 1 (13); ARS1211: 2 (19); ARS1212: 4 (35); ARS1212.5: 7 (7); ARS1213: 3 (31); ARS1214: 2 (2); ARS1215: 3 (27); ARS1216: 20 (164); ARS1217: 9 (73); ARS1218: 7 (59); ARS1219: 1 (10); ARS1220: 16 (131); ARS1221: 2 (2); ARS1223: 5 (46); ARS1226: 1 (15); ARS1228: 2 (2); ARS1230: 2 (2); ARS1232: 2 (21); ARS1233: 7 (7); ARS1234: 8 (65); ARS1235: 7 (57); ARS1238: 2 (17); ARS1303: 1 (14); ARS1304: 2 (19); ARS1305: 1 (8); ARS1307: 5 (44); ARS1307.5: 2 (23); ARS1308: 1 (8); ARS1309: 9 (72); ARS1310: 6 (54); ARS1312: 2 (23); ARS1316: 1 (14); ARS1317: 2 (2); ARS1319: 5 (43); ARS1320: 8 (70); ARS1321: 2 (2); ARS1322: 2 (17); ARS1323: 3 (29); ARS1324: 5 (45); ARS1325: 2 (18); ARS1327: 5 (47); ARS1328: 5 (42); ARS1329: 1 (15); ARS1330: 1 (9); ARS1331: 1 (9); ARS1332: 8 (71); ARS1333: 1 (15); ARS1405: 5 (47); ARS1406: 5 (44); ARS1407: 8 (70); ARS1409: 2 (2); ARS1410: 2 (20); ARS1411: 7 (62); ARS1412: 5 (40); ARS1413: 3 (24); ARS1414: 2 (18); ARS1415: 5 (45); ARS1416: 3 (3); ARS1417: 5 (44); ARS1419: 3 (31); ARS1420: 10 (83); ARS1421: 1 (15); ARS1422: 3 (27); ARS1423: 1 (1); ARS1424: 3 (29); ARS1425: 1 (1); ARS1426: 1 (11); ARS1427: 6 (53); ARS1501: 2 (21); ARS1502: 5 (46); ARS1507: 4 (36); ARS1508: 1 (12); ARS1509: 4 (32); ARS1510: 3 (24); ARS1511: 7 (59); ARS1512: 5 (42); ARS1513: 2 (18); ARS1516: 2 (20); ARS1518: 2 (18); ARS1519: 4 (33); ARS1521: 6 (54); ARS1523: 2 (16); ARS1524: 2 (21); ARS1526: 2 (19); ARS1528: 5 (40); ARS1529: 3 (28); ARS1531: 6 (50); ARS1603: 4 (4); ARS1604: 2 (16); ARS1605: 4 (34); ARS1607: 6 (51); ARS1608: 15 (121); ARS1611: 1 (1); ARS1612: 3 (31); ARS1614: 7 (56); ARS1617: 3 (27); ARS1618: 4 (37); ARS1619: 3 (26); ARS1620: 2 (2); ARS1621: 5 (46); ARS1622: 7 (57); ARS1623: 5 (41); ARS1624: 1 (11); ARS1625: 12 (101); ARS1626: 2 (16); ARS1627: 6 (49); ARS1628: 2 (16); ARS1629: 3 (3); ARS1630: 4 (35); ARS1631: 2 (23); ARS1632: 8 (71); ARS1633: 5 (46); ARS1634: 7 (57); ARS1635: 8 (65); ARS201: 12 (103); ARS202: 5 (45); ARS203: 4 (32); ARS206: 5 (47); ARS207: 5 (47); ARS208: 3 (30); ARS209: 5 (5); ARS211: 4 (33); ARS212: 7 (7); ARS213: 14 (116); ARS214: 3 (26); ARS215: 1 (14); ARS216: 10 (83); ARS217: 1 (8); ARS220: 1 (10); ARS221: 1 (9); ARS222: 22 (177); ARS224: 15 (122); ARS225: 6 (50); ARS228: 5 (45); ARS230: 9 (78); ARS231: 1 (8); ARS301: 1 (1); ARS302: 4 (4); ARS303: 4 (4); ARS304: 1 (9); ARS305: 1 (8); ARS306: 3 (25); ARS307: 4 (33); ARS308: 1 (14); ARS309: 3 (31); ARS310: 4 (37); ARS313: 2 (19); ARS314: 4 (38); ARS315: 30 (246); ARS316: 3 (30); ARS317: 2 (20); ARS318: 7 (7); ARS320: 4 (4); ARS403: 4 (33); ARS404: 6 (52); ARS405: 7 (57); ARS406: 1 (8); ARS409: 5 (40); ARS409.5: 2 (2); ARS410: 9 (72); ARS411: 3 (3); ARS412: 4 (32); ARS413: 9 (78); ARS414: 1 (14); ARS415: 2 (23); ARS416: 14 (119); ARS417: 29 (238); ARS418: 5 (40); ARS419: 12 (102); ARS420: 1 (12); ARS421: 3 (27); ARS422: 3 (26); ARS423: 3 (26); ARS425: 7 (61); ARS427: 1 (8); ARS428: 10 (87); ARS429: 7 (7); ARS430: 29 (237); ARS431: 2 (17); ARS432: 48 (388); ARS433: 5 (43); ARS434: 1 (15); ARS435: 3 (31); ARS436: 2 (2); ARS439: 1 (10); ARS440: 4 (32); ARS442: 2 (2);

ARS443: 3 (28); ARS446: 5 (43); ARS450: 9 (72); ARS451: 1 (14); ARS452: 5 (42); ARS453: 5 (40); ARS502: 2 (2); ARS503: 3 (24); ARS504: 2 (2); ARS507: 9 (72); ARS508: 7 (57); ARS510: 6 (51); ARS511: 4 (39); ARS512: 2 (23); ARS513: 2 (2); ARS513.5: 5 (5); ARS513.7: 1 (14); ARS514: 8 (65); ARS515: 1 (11); ARS516: 2 (17); ARS517: 8 (66); ARS518: 12 (103); ARS519: 2 (2); ARS520: 3 (26); ARS521: 3 (3); ARS522: 2 (20); ARS523: 2 (21); ARS600: 3 (28); ARS600.4: 16 (128); ARS601: 3 (28); ARS602: 3 (28); ARS603: 1 (10); ARS603.1: 2 (2); ARS603.5: 5 (46); ARS604: 2 (2); ARS605: 18 (146); ARS606: 4 (39); ARS607: 10 (81); ARS608: 1 (15); ARS609: 4 (37); ARS701: 2 (2); ARS702: 3 (29); ARS704: 3 (26); ARS706: 4 (4); ARS707: 2 (21); ARS709: 4 (4); ARS710: 2 (21); ARS712: 2 (21); ARS714: 4 (36); ARS715: 3 (3); ARS716: 17 (142); ARS717: 1 (8); ARS718: 3 (29); ARS719: 6 (54); ARS720: 11 (95); ARS721: 41 (335); ARS722: 1 (9); ARS723: 1 (9); ARS724: 2 (2); ARS727: 2 (19); ARS728: 4 (34); ARS729: 5 (40); ARS731: 4 (34); ARS733: 1 (12); ARS734: 8 (69); ARS735: 2 (2); ARS736: 6 (54); ARS737: 7 (63); ARS801: 4 (37); ARS802: 2 (17); ARS805: 1 (12); ARS806: 2 (20); ARS807: 9 (72); ARS808: 1 (1); ARS809: 3 (24); ARS810: 21 (172); ARS811: 28 (231); ARS813: 7 (62); ARS814: 2 (2); ARS815: 10 (81); ARS818: 5 (47); ARS820: 7 (60); ARS822: 3 (27); ARS824: 7 (56); ARS902: 7 (62); ARS904: 1 (13); ARS907: 3 (29); ARS909: 9 (75); ARS910: 2 (2); ARS911: 2 (16); ARS912: 3 (27); ARS913: 19 (157); ARS914: 5 (44); ARS915: 5 (44); ARS918: 2 (2); ARS919: 2 (21); ARS920: 2 (17); ARS921: 1 (1); ARS922: 8 (67); ARS923: 2 (16); ARS1002: 60 (481); ARS1003: 5 (40); ARS1004: 16 (135); ARS1005: 29 (238); ARS1006: 5 (44); ARS1007: 18 (148); ARS1008: 12 (97); ARS1009: 12 (97); ARS1010: 7 (60); ARS1011: 33 (265); ARS1012: 64 (517); ARS1013: 64 (517); ARS1014: 8 (68); ARS1015: 11 (94); ARS1016: 7 (7); ARS1017: 3 (29); ARS1018: 21 (171); ARS1019: 16 (133); ARS1020: 14 (115); ARS1021: 5 (43); ARS1022: 10 (83); ARS1023: 11 (90); ARS1024: 6 (53); ARS103: 3 (24); ARS104: 7 (60); ARS105: 25 (207); ARS106: 4 (32); ARS107: 19 (159); ARS108: 5 (47); ARS109: 25 (201); ARS110: 6 (52); ARS1102: 3 (3); ARS1103: 85 (681); ARS1104: 3 (3); ARS1106: 9 (79); ARS1109: 39 (315); ARS111: 42 (339); ARS1112: 28 (230); ARS1113: 27 (223); ARS1114: 14 (115); ARS1115: 15 (126); ARS1116: 10 (85); ARS1118: 10 (84); ARS112: 2 (23); ARS1120: 3 (28); ARS1123: 50 (407); ARS1125: 11 (89); ARS1126: 1 (12); ARS1127: 8 (65); ARS1200-1: 3010 (24080); ARS1200-2: 1348 (10784); ARS1202: 31 (250); ARS1206: 6 (49); ARS1207: 36 (292); ARS1208: 1 (12); ARS1209: 16 (128); ARS1210: 5 (43); ARS1211: 14 (113); ARS1212: 17 (140); ARS1212.5: 1 (15); ARS1213: 13 (104); ARS1214: 3 (3); ARS1215: 7 (60); ARS1216: 39 (319); ARS1217: 18 (151); ARS1218: 6 (53); ARS1219: 2 (16); ARS1220: 25 (201); ARS1221: 3 (3); ARS1223: 5 (47); ARS1226: 7 (57); ARS1228: 3 (3); ARS1230: 3 (3); ARS1232: 8 (67); ARS1233: 5 (47); ARS1234: 16 (134); ARS1235: 10 (84); ARS1238: 4 (34); ARS1303: 6 (48); ARS1304: 5 (44); ARS1305: 2 (20); ARS1307: 4 (38); ARS1307.5: 5 (47); ARS1308: 3 (27); ARS1309: 21 (173); ARS1310: 22 (176); ARS1312: 6 (49); ARS1316: 3 (24); ARS1317: 2 (2); ARS1319: 12 (98); ARS1320: 7 (57); ARS1321: 3 (3); ARS1322: 7 (56); ARS1323: 4 (39); ARS1324: 16 (131); ARS1325: 2 (22); ARS1327: 14 (115); ARS1328: 30 (245); ARS1329: 5 (41); ARS1330: 3 (31); ARS1331: 2 (22); ARS1332: 33 (265); ARS1333: 5 (40); ARS1405: 41 (329); ARS1406: 10 (81); ARS1407: 26 (213); ARS1409: 3 (3); ARS1410: 11 (92); ARS1411: 30 (242); ARS1412: 17 (139); ARS1413: 15 (121); ARS1414: 8 (70); ARS1415: 13 (108); ARS1416: 2 (2); ARS1417: 15 (123); ARS1419: 11 (89); ARS1420: 17 (139); ARS1421: 5 (45); ARS1422: 7 (61); ARS1423: 3 (3); ARS1424: 9 (75); ARS1425: 3 (3); ARS1426: 2 (22); ARS1427: 18 (146); ARS1501: 4 (38); ARS1502: 10 (80); ARS1507: 7 (62); ARS1508: 5 (47); ARS1509: 5 (47); ARS1510: 5 (42); ARS1511: 23 (186); ARS1512: 6 (53); ARS1513: 5 (41); ARS1516: 12 (98); ARS1518: 10 (84); ARS1519: 9 (72); ARS1521: 18 (151); ARS1523: 6 (48); ARS1524: 4 (38); ARS1526: 4 (33); ARS1528: 26 (210); ARS1529: 11 (95); ARS1531: 7 (59); ARS1603: 1 (8); ARS1604: 4 (34); ARS1605: 15 (120); ARS1607: 20 (167); ARS1608: 7 (60); ARS1611: 2 (2); ARS1612: 14 (114); ARS1614: 11 (92); ARS1617: 2 (19); ARS1618: 21 (171); ARS1619: 5 (42); ARS1620: 4 (4); ARS1621: 11 (91);

ARS1622: 33 (270); ARS1623: 23 (189); ARS1624: 2 (22); ARS1625: 21 (171); ARS1626: 5 (41); ARS1627: 11 (88); ARS1628: 7 (62); ARS1629: 7 (7); ARS1630: 5 (40); ARS1631: 10 (84); ARS1632: 27 (217); ARS1633: 5 (41); ARS1634: 33 (270); ARS1635: 43 (347); ARS201: 39 (319); ARS202: 5 (44); ARS203: 2 (23); ARS206: 53 (428); ARS207: 29 (239); ARS208: 10 (87); ARS209: 2 (18); ARS211: 23 (185); ARS212: 2 (16); ARS213: 60 (483); ARS214: 5 (44); ARS215: 6 (53); ARS216: 36 (291); ARS217: 3 (28); ARS220: 5 (44); ARS221: 2 (21); ARS222: 69 (552); ARS224: 60 (483); ARS225: 20 (160); ARS228: 35 (282); ARS230: 52 (416); ARS231: 2 (21); ARS301: 2 (2); ARS302: 1 (12); ARS303: 1 (12); ARS304: 2 (19); ARS305: 2 (20); ARS306: 9 (72); ARS307: 11 (91); ARS308: 4 (39); ARS309: 9 (75); ARS310: 34 (277); ARS313: 25 (201); ARS314: 9 (76); ARS315: 49 (393); ARS316: 6 (49); ARS317: 4 (33); ARS318: 6 (51); ARS320: 1 (12); ARS403: 12 (101); ARS404: 21 (171); ARS405: 11 (91); ARS406: 4 (33); ARS409: 22 (176); ARS409.5: 4 (4); ARS410: 70 (562); ARS411: 3 (3); ARS412: 5 (46); ARS413: 12 (97); ARS414: 4 (35); ARS415: 7 (63); ARS416: 20 (162); ARS417: 82 (662); ARS418: 8 (69); ARS419: 4 (37); ARS420: 3 (28); ARS421: 2 (22); ARS422: 15 (125); ARS423: 4 (32); ARS425: 12 (101); ARS427: 4 (34); ARS428: 20 (160); ARS429: 3 (24); ARS430: 77 (622); ARS431: 10 (83); ARS432: 116 (930); ARS433: 83 (664); ARS434: 4 (35); ARS435: 6 (48); ARS436: 2 (2); ARS439: 1 (13); ARS440: 3 (28); ARS442: 3 (3); ARS443: 13 (109); ARS446: 17 (140); ARS450: 10 (87); ARS451: 2 (22); ARS452: 7 (63); ARS453: 5 (45); ARS502: 3 (3); ARS503: 10 (80); ARS504: 3 (3); ARS507: 19 (159); ARS508: 23 (185); ARS510: 23 (190); ARS511: 14 (112); ARS512: 18 (150); ARS513: 3 (3); ARS513.5: 2 (19); ARS513.7: 4 (37); ARS514: 10 (83); ARS515: 3 (30); ARS516: 7 (62); ARS517: 7 (56); ARS518: 24 (192); ARS519: 3 (3); ARS520: 21 (168); ARS521: 6 (6); ARS522: 8 (69); ARS523: 27 (219); ARS600: 30 (246); ARS600.4: 53 (431); ARS601: 10 (82); ARS602: 10 (82); ARS603: 3 (29); ARS603.1: 7 (7); ARS603.5: 5 (47); ARS604: 5 (5); ARS605: 30 (247); ARS606: 5 (45); ARS607: 12 (96); ARS608: 4 (38); ARS609: 11 (91); ARS701: 3 (3); ARS702: 10 (80); ARS704: 5 (40); ARS706: 6 (6); ARS707: 9 (77); ARS709: 2 (16); ARS710: 7 (63); ARS712: 8 (66); ARS714: 13 (104); ARS715: 6 (6); ARS716: 25 (207); ARS717: 2 (20); ARS718: 12 (100); ARS719: 20 (162); ARS720: 5 (43); ARS721: 101 (809); ARS722: 3 (27); ARS723: 3 (27); ARS724: 3 (3); ARS727: 7 (60); ARS728: 6 (51); ARS729: 17 (137); ARS731: 3 (28); ARS733: 7 (61); ARS734: 25 (204); ARS735: 1 (10); ARS736: 15 (126); ARS737: 24 (197); ARS801: 24 (193); ARS802: 8 (70); ARS805: 4 (39); ARS806: 7 (63); ARS807: 16 (131); ARS808: 4 (4); ARS809: 6 (54); ARS810: 190 (1522); ARS811: 362 (2902); ARS813: 23 (184); ARS814: 3 (3); ARS815: 50 (402); ARS818: 16 (132); ARS820: 16 (134); ARS822: 6 (51); ARS824: 25 (205); ARS902: 32 (257); ARS904: 6 (49); ARS907: 19 (152); ARS909: 24 (197); ARS910: 5 (5); ARS911: 7 (59); ARS912: 21 (168); ARS913: 22 (180); ARS914: 18 (148); ARS915: 18 (148); ARS918: 3 (3); ARS919: 9 (72); ARS920: 7 (56); ARS921: 2 (2); ARS922: 12 (102); ARS923: 8 (66).

(PDF)

**S5 Fig. Mcm2-, Mcm4-, and Mcm6-ChEC footprints over 12 origins on chrIV.** Footprints for Mcm2-ChEC (top row), Mcm4-ChEC (second row), and Mcm6-ChEC (third row) resemble each other at each of 12 consecutive origins on chrIV, but free MNase (bottom row) does not. Below each heat map is a relative distribution of read depths for the 50–100 bp size range. Capped and uncapped read depths for MCM2 are described and listed in the legend to S4 Fig. Corresponding numbers for MCM4 and MCM6 are listed below. MCM4 ARS429: 7 (7); MCM4 ARS430: 29 (237); MCM4 ARS431: 2 (17); MCM4 ARS432: 48 (388); MCM4 ARS433: 5 (43); MCM4 ARS434: 1 (15); MCM4 ARS435: 3 (31); MCM4 ARS436: 2 (2); MCM4 ARS439: 1 (10); MCM4 ARS440: 4 (32); MCM4 ARS452: 5 (42); MCM4 ARS453: 5 (40); MCM6 ARS429: 3 (24); MCM6 ARS430: 77 (622); MCM6 ARS431: 10 (83); MCM6 ARS432: 116 (930); MCM6 ARS433: 83 (664); MCM6 ARS434: 4 (35); MCM6 ARS435: 6 (48); MCM6

ARS436: 2 (2); MCM6 ARS439: 1 (13); MCM6 ARS440: 3 (28); MCM6 ARS452: 7 (63); MCM6 ARS453: 5 (45).
(TIF)

**S6 Fig. Multiple MCM signals do not necessarily reflect multiple MCM binding events in the same cell.** The heat plots that we use to visualize MCM binding in Figs 1B, 2A, 3A, S4 and S6 often show more than one adjacent MCM footprints. Because these plots reflect the composite binding of all of the cells in the population, the presence of multiple MCM signals need not indicate that more than one MCM DH binds within the same cell. Multiple MCM binding events within the same cell should generate fragments that span the region between those sites (red fragment in the top image). In the plot shown (ARS1411), this fragment should be approximately 75 base pairs long, as indicated by its position on the y axis; its absence (space between two red arrows) indicates that the two MCM DH footprints at this origin mostly reflect signals that emanate from two populations of cells, each with a single MCM DH. However, we cannot exclude the possibility that the 75 bb fragment could have been degraded. The single vertical dotted yellow line indicates the position of the ACS. The pairs of vertical gray and green lines delineate the ARS boundaries, as reported in SGD, and 60 base pairs on either side of the peak of Mcm2-ChEC signal.
(TIF)

**S7 Fig. Fragment size distributions for Mcm2-MNase libraries prepared from fission yeast.**
(TIF)

**S8 Fig. All "confirmed" *S. pombe* origins from OriDB presented as heat maps.** Each figure shows a 3 kb span on the X axis. Fragment sizes range from 0 to 200 bp on the Y axis. Plots are centered on midpoints of origins, as listed in OriDB. Read depths are listed as described in S4 Fig: chrIII_120001 1 (10); chrIII_123404 6 (6); chrIII_1411610 1 (8); chrIII_1417150 6 (6); chrIII_1719145 6 (6); chrIII_1823925 6 (6); chrIII_1838655 2 (20); chrIII_1844225 1 (10); chrIII_3479 259 (2074); chrII_1254200 1 (10); chrII_1278060 1 (9); chrII_1547425 5 (5); chrII_1625285 3 (29); chrII_2105805 3 (30); chrII_2118500 3 (25); chrII_2187470 1 (15); chrII_24041 20 (160); chrII_2582435 3 (31); chrII_2790759 6 (52); chrII_2800137 2 (21); chrII_3154425 35 (281); chrII_31591 1 (14); chrII_3334855 1 (9); chrII_3344900 5 (5); chrII_3357660 2 (18); chrII_3362660 1 (10); chrII_3374145 6 (6); chrII_3398460 6 (6); chrII_4141200 5 (44); chrII_448341 4 (32); chrII_603722 3 (26); chrI_1028000 3 (24); chrI_1094953 3 (24); chrI_113108 2 (16); chrI_1149805 1 (8); chrI_1252175 5 (5); chrI_2225555 4 (4); chrI_2438581 12 (103); chrI_3059510 4 (33); chrI_3247323 3 (29); chrI_327080 5 (46); chrI_3500690 4 (4); chrI_3954285 1 (9); chrI_4072925 1 (9); chrI_4254830 2 (19); chrI_5025445 4 (33); chrI_5046440 6 (55); chrI_712465 6 (6).
(TIF)

**S9 Fig. Fragment sizes for MCM2-ChEC libraries prepared from either *S. cerevisiae* or *S. pombe* are consistent between cultures that have been arrested in G1 or using HU.** *S. cerevisiae* ARS1103 is a highly active origin that replicates early in S-phase.
(TIF)

**S10 Fig. Replica pombe Mcm2-ChEC measurements at replication initiation and termination sites, as determined by Pu-seq [47].** A. Pu-seq (red and blue for pol epsilon and pol delta, respecively) and Mcm2-ChEC (black) signal over 150 kb stretch of chrII. Replicate Mcm2-ChEC measurements (top two rows) show similar colocalization with sites of replication initiation, but not with sites of replication termination, as described in the text. Free MNase (bottom row) does not show colocalization with replication initiation or termination

sites. All fragment sizes are included in the analysis. B. Genome-wide quantitation of data from A expressed as violin plots.
(TIF)

**S11 Fig. Immunoblot showing expression level of MCM-MNase fusion protein in HeLa and mouse hybrid *M.musculus/M.castaneus* Patski cells.** The relative abundance of tagged vs untagged bands for human and mouse cells was 0.4- and 0.25- fold, respectively.
(TIF)

**S12 Fig. Size distribution of Mcm2-ChEC fragments in all four organisms examined.** A. Size distribution of Mcm2-ChEC library fragments in HeLa cells. B. Size distribution of Mcm2-ChEC library fragments in budding yeas (black), fission yeast (blue), mouse (red) and human (purple).
(TIF)

**S13 Fig. Replicate Mcm2-ChEC measurements in HeLa cells, binned in 10 kb bins, are highly correlated (r = 0.881) with each other, but not with free MNase (r = 0.199).** All fragment sizes are included in the analysis.
(TIF)

**S14 Fig.  A-F.** Six examples of replication fork direction assays showing fraction of forks that synthesize the Watson strand that are moving rightward (red and blue dots used to indicate direction of movement of majority of forks) juxtaposed with MCM binding (black) as in Fig 7A. All Mcm2-ChEC fragment sizes are included in the analysis.
(PDF)

**S15 Fig. Replicate measurements of Mcm2-ChEC in HeLa show similar colocalization with replication initiation sites, as determined by Okazaki-seq.** A. Dots represent the fraction of forks involved in synthesis of the Watson strand that are moving to the right, as described in Fig 7. Dots are red when >50% of forks are moving rightward, and blue otherwise. Replicate measurements of Mcm2-ChEC (black lines in two top panels) show comparable colocalization with replication initiation sites, but free MNase (black line in bottom left panel) does not. See text and Fig 7 for details. B. Quantitation of free MNase results from A, illustrated as a violin plot. "NS" indicates that the difference between levels of free MNase/G1 DNA ratio signal at replication initiation and termination sites are not significantly different.
(TIF)

**S16 Fig. Size distribution and reproducibility of Mcm2-ChEC in mouse.** A. Size distribution of Mcm2-ChEC library. B. Replicate measurements of Mcm2-ChEC signal (10 kb bins) are highly correlated (R = 0.96).
(TIF)

**S17 Fig. Difference in replication timing between the two homologs is most extreme for X chromosome.** Absolute Deviations (MAD) of read depths in S-phase along M. spretus and M. musculus chromosomes were used to measure replication fluctuations for each chromosome. The ratio of fluctuations between two chromosomes was highest for the X chromosome (2.8), indicating that the difference in replication time between the two homologs is most extreme for this chromosome. Chromosomes are arranged in order of fluctuation ratios.
(TIF)

**S18 Fig. MCM abundance does not correlated with replication timing across OriDB origins as determined by Yabuki et al. [50], but it correlates with replication activity in HU as determined by ssDNA across the origins [46].** A. Mcm2-ChEC signal at 829 S. cerevisiae origins listed in OriDB is not correlated with replication timing. B. Replication activity at OriDB

origins excluding repetitive regions, as measured by single-stranded DNA [46] is correlated with both Mcm2-ChEC (R = 0.65) and Mcm-ChIP (R = 0.57) [45] signal. All Mcm2-ChEC fragment sizes are included in the analysis.
(TIF)

**S1 Table.** *S. cerevisiae* **ARSs with fractions of MCM signal that arise from single peaks (linked Excel file).**
(XLSX)

**S2 Table. Mcm-ChEC versus published Mcm-ChIP (linked Excel file).**
(XLSX)

**S3 Table. List of yeast strains.** (Word document).
(DOCX)

**S4 Table. Sequence quality information (linked Excel file).**
(XLSX)

**S1 Source Files. Zip file containing the following: data underlying Figs 1A and 1B with a code example for generating heat maps, C; 2A,B,C; 3B; 4A,B; 5A,B,C; 6A,B,C,D; 7A.**
(ZIP)

**S2 Source Files. Zip file containing the following: data underlying Fig 7B part 1 of 2.**
(ZIP)

**S3 Source Files. Zip file containing the following: data underlying Figs 7B part 2 of 2.**
(ZIP)

**S4 Source Files. Zip file containing the following: data underlying Figs 8; 9A, 9B; S2; S3.**
(ZIP)

**S5 Source Files. Zip file containing the following: data underlying S4 Fig traces.**
(ZIP)

**S6 Source Files. Zip file containing the following: data underlying S4 Fig heat maps.**
(ZIP)

**S7 Source Files. Zip file containing the following: data underlying S5, S6, S7, S8, S9, S10A and S10B; S12A and S12B Figs.**
(ZIP)

**S8 Source Files. Zip file containing the following: data underlying S13A and S13B Figs.**
(ZIP)

**S9 Source Files. Zip file containing the following: data underlying Figs S14A,14B,14C,14D,14E and 14F; S15A.**
(ZIP)

**S10 Source Files. Zip file containing the following: data underlying S15B Fig part 1 of 2.**
(ZIP)

**S11 Source Files. Zip file containing the following: data underlying S15B Fig part 2 of 2.**
(ZIP)

**S12 Source Files. Zip file containing the following: data underlying Figs S16A and 16B; S17; S18A and 18B.**
(ZIP)

## Acknowledgments

We are grateful to the ENCODE Consortium and the laboratories of John Stamatoyannopoulos and Bradley Bernstein for generating the repli-seq and histone modification data sets for GM12878 and HeLa cell lines. We thank Hervé Pagès for computational advice and Olivier Hyrien for sharing data.

## Author Contributions

**Conceptualization:** Eric J. Foss, Smitha Sripathy, Tonibelle Gatbonton-Schwager, Antonio Bedalov.

**Data curation:** Eric J. Foss.

**Formal analysis:** Eric J. Foss, Antonio Bedalov.

**Funding acquisition:** Antonio Bedalov.

**Investigation:** Eric J. Foss, Smitha Sripathy, Tonibelle Gatbonton-Schwager, Hyunchang Kwak, Adam H. Thiesen, Uyen Lao, Antonio Bedalov.

**Methodology:** Eric J. Foss, Smitha Sripathy, Tonibelle Gatbonton-Schwager, Hyunchang Kwak, Adam H. Thiesen, Uyen Lao, Antonio Bedalov.

**Project administration:** Antonio Bedalov.

**Resources:** Antonio Bedalov.

**Software:** Eric J. Foss.

**Supervision:** Uyen Lao, Antonio Bedalov.

**Writing – original draft:** Eric J. Foss, Antonio Bedalov.

**Writing – review & editing:** Eric J. Foss, Smitha Sripathy, Tonibelle Gatbonton-Schwager, Antonio Bedalov.

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
