## [Decision Letter · Decision Letter 0]

8 Dec 2020

Dear Tony,

Thank you very much for submitting your Research Article entitled 'Chromosomal Mcm2-7 distribution is the primary driver of the genome replication program in species from yeast to humans' to PLOS Genetics. Also, thank you for making the additional information requested available to reviewers. I regret the extended period that the paper was out for review, mainly due to a delay in receiving one of the reviews.

The manuscript was fully evaluated at the editorial level and by independent peer reviewers. The reviewers appreciated the attention to an important problem, but raised substantial concerns about the current manuscript. Based on the reviews, we will not be able to accept this version of the manuscript, but we would be willing to review a much-revised version. We cannot, of course, promise publication at that time.

Below are some specific comments that I (ML) have, which I hope will be helpful in crafting your revision and response. You need not respond to them specifically since they are also comments made by the reviewers (to which you should respond, of course).

1. There was a general feeling that the paper would benefit from more in-depth analysis and discussion; to me the paper feels as if it had been originally written with page/word limits in mind. This should not be a concern with an online journal like PLOS Genetics. The Materials and Methods section is particularly brief.

2. There was also a concern that the ChEC-Seq approach was not sufficiently validated, particularly in S. pombe and in mammalian cells. To reiterate one reviewer's comment, if high-efficiency origins have been identified in these cells (where admittedly most are not), perhaps those could be used for validation.

3. There was also a concern that the tagged Mcm constructs in budding yeast were not fully functional. I don't think colony size is sufficient. Since you used FACS-sorting to isolate G1 and S phase populations, it should be relatively straightforward to quantify the fraction of cells in G1, S and G2 in tagged and untagged strains as a test.

If you decide to revise the manuscript for further consideration at PLOS Genetics, please aim to resubmit within the next 60 days, unless it will take extra time to address the concerns of the reviewers, in which case we would appreciate an expected resubmission date by email to plosgenetics@plos.org.

[LINK]

We are sorry that we cannot be more positive about your manuscript at this stage. Please do not hesitate to contact us if you have any concerns or questions.

Yours sincerely,

Michael Lichten, Ph.D.

Associate Editor

PLOS Genetics

Gregory P. Copenhaver

Editor-in-Chief

PLOS Genetics

Reviewer's Responses to Questions

**Comments to the Authors:**

Reviewer #1: Review has been uploaded as an attachment.

Reviewer #2: Foss et al. present a new and powerful approach for mapping MCM complexes in G1 cells. Using CheC-seq, which tethers MNase to either side of the complex and releases MCM footprints upon activation, they map MCM at near nucleotide resolution in budding yeast, fission yeast, mouse and human. Based on similarities between MCM-loading patterns and replication timing profiles, and also based on their modeling, they conclude that stochastic activation of MCMs is sufficient to explain replication kinetics. This result is an important addition to a growing body of literature supporting stochastic models of replication timing. As such, it is a significant contribution to the field, and one that nicely spans yeast and vertebrates. Having said that, I am a bit disappointed in the paper, both in the way the data is analyzed and in the way the analyses are interpreted. My disappointment with the analyses is that they overly anecdotal—showing examples of data instead of doing rigorous genome-wide analyses. My disappointment with the interpretation is that the authors push two simplistic model even though there data suggests more complexed realities.

The first model is that stochastic firing of MCM explains "most differences in replication timing". Others have argued that MCM distribution is an important parameter, but it is clear other factors affect timing. The authors themselves admit as much, but in away that suggests MCM distribution is all that matters and that other factors should be discounted. However, their own best evidence for the point (Figure S12, which should be moved to the primary figures) shows at best r = 0.72, implying that on Chromosome IV, MCM distribution explains about half (0.72^2 = 0.52) of the variance in timing. What about the chromosomes they don't show? What is the correlation coefficient for the whole genome? Or perhaps the whole euchromatic genome? The analysis for the human genome is even more anecdotal, analyzing only 80 Mb.

The second model is that there is only one MCM-DH loaded at any budding yeast origin. Again, the data shows that it is more complicated and the authors admit as much, but argue strongly that one is all that you need and that origins with multiple MCMs, if the actually exist, are not relevant. The motivation for their vehemence on this point is unclear, because their main point—that MCM distribution determines timing patterns—is unaffected by absolute MCM stoichiometry.

It is the authors prerogative to push their interpretations as hard as they like. However, I suspect they would convince more of their readers, and probably end up being closer to correct, if they would be a bit more nuanced in their arguments.

In any case, addressing the following points would also strengthen their case.

They use their modeling to conclude that, in budding yeast, MCM distribution explains replication timing. They should do the same thing with the actual MCM distribution. They should plot MCM signal versus replication timing at each origin.

They should show the distribution of single-MCM-peak versus multiple-MCM-peak origins, either as a histogram of fraction of MCM signal arising from a single DH or as a scatter plot of single-MCM% versus total MCM signal.

Their estimation that origins with multiple MCM peaks comprise only 25% of the total MCM signal is skewed by inclusion of the two rDNA origins, which (as they are repeated about 100 times) comprise 35% of the total signal. Excluding them (or including them at normalized levels), the one MCM/multiple MCM ratio is about 60/40.

They make a strong case that less than 1000 MCMs (500 MCM-DHs) are loaded each G1 in budding yeast, yet numerous studies (e.g., https://www.ncbi.nlm.nih.gov/pubmed/29361465 ) have estimated that there are 3000 to 5000 MCMs per cell, most of them loaded during G1. The authors need to address this discrepancy.

They say that they "used the same approach" for pombe as cerevisiae, but the method of synchronization, HU instead of mating pheromone, is substantially different, and should be noted and explained. The fact that the HU and G1 data sets look similar also needs explaining because, by their model, the efficient origins—such as ARS1103, which fires is >80% of HU-arrested cells—should be almost completely depleted of MCM-DHs is the HU data set.

They should site the literature that describes stochastic models similar to theirs and likewise concludes that stochastic activation of MCM complexes is sufficient to explain replication timing patterns, in particular d'Moura https://pubmed.ncbi.nlm.nih.gov/20457753 , Yang https://www.ncbi.nlm.nih.gov/pubmed/20739926 and Gindin https://www.ncbi.nlm.nih.gov/pubmed/24682507 .

I would be very interested to know how efficient the ChEC cleavage is. If it is efficient, it would strengthen their claim that they are mapping every loaded MCM. However, if it is inefficient, it is possible that they are detecting a biased subset of MCMs. A simple genomic Southern blot before and after calcium treatment would suffice.

They should include in a supplemental table information about the size and quality of their sequence data sets.

Minor Issues:

ACS is 17 bp, not 11 bp. Both cited papers show a 17 bp ASC.

On page 16, the phrase "no difference" is a bit strong; "no significant difference" would be more accurate, and significant should be defined.

The red arrow in Figure 1B is presumably the ACS. If so, it should be so labeled.

The genomic coordinates of the regions shown in Figure 2A should be provided.

The yeast replication profile data in Figure 2A should be cited.

The y axes of the bottom rows of Figure 2A should be properly labeled. The y values of these graphs are almost certainly not simply read depth.

The library read-size distribution for all species should be plotted on the same graph for easier comparison.

Reporting a p value of 5.3x10E-46, and the implication that there is a meaningful difference between 10E-46 and 10E-63, is silly. p < 10E-10 would suffice and model a more realistic approach to statistics.

The authors claim that it "is clear ... the number of early- and late-replicating regions on a single chromosome is similar". Absent quantitative, genome-wide analysis, it is not clear. Furthermore, it is immaterial to the paper and should be removed.

Reviewer #3: Review of Foss, Sripathy, et al. (Bedalov) “Chromosomal MCM2-7 distribution is the primary driver of the genome replication program in species from yeast to humans”

MCM complex binding to chromosomes has been assessed in multiple independent experiments over the past decades in budding yeast using genome-scale Chip-chip and MCM ChIPSeq approaches. In yeast, distinct MCM ChIP signals corresponding to known origins of replication can be easily detected and closely overlap ORC binding sites, consistent with the finding that in this organism origins are specified by discrete DNA regions of less than 200 bp. In contrast, in other organisms, particular metazoans including human cells, MCM complex binding sites have been much more difficult to map precisely. In Drosophila cells, the MacAlpine lab showed that the cyclin E/Cdk2 promotes multiple MCM complex loading events and that the loaded MCM complexes are then distributed over chromosomes at least in part by active transcription (Powell et al, EMBO J 2015). Thus every single cell in a clonal population potentially possesses a distinct distribution of MCM complexes, making discrete MCM complex sites virtually impossible to map. However, in the absence of E/Cdk2, 10-fold fewer MCM complexes are loaded and these are localized to known ORC sites, as occurs in yeast. Regardless, the relationship between MCM complex levels, distribution and replication time has been challenging to tease out in eukaryotes, including yeast. Moreover, in yeast there is strong published evidence that the localization of limiting S-phase factors (i.e. the ability of these S-phase factors to access licensed origins) plays a primary role in controlling origin activation time, and thus the replication timing ‘program’, which runs counter to the major conclusion that the authors of this study are making, at least based on the title and abstract sections of their paper (they do acknowledge limitations or exceptions to these conclusions in their Discussion)

In this study, the authors adapt a relatively new genome-scale protein mapping procedure developed in by the Henikoff lab called ChEC (chromatin endogenous cleavage) to map MCM binding sites in budding yeast, fission yeast and mammalian cells. In these experiments individual Mcm subunits are fused to MNase; cells are isolated between MG1, the period where MCM complexes are loaded, and permeabilzed. The addition of CaCl2 activates the MNase which cleaves DNA where MCM complexes are bound. The released fragments sequenced and mapped to the genome. Fragments of ~ 60 bp, the approximate region protected by an MCM complex, are interpreted as loaded MCM complexes.

Using this method across the cells of different species, the authors make the general conclusion that, despite the differences in molecular determinants of origins and ORC binding mechanisms (e.g. a discrete conserved DNA sequence is used by budding yeast ORC, an AT-rich region and a species specific AT-hook on one Orc subunit is used by S.pombe ORC, no simple sequence rule is used by mammalian ORC, but it appears to often accumulate in “open chromatin” regions) the distribution of loaded MCM complexes determines/correlates with the replication timing pattern of the genome, with regions containing higher densities of MCM signals replicating earlier and those containing lower densities replicating later. Thus they conclude that the distribution of licensed origins is a primary driver of the genome replication program.

While there are a lot of interesting and useful new data in this study, no single line of inquiry was developed sufficiently enough to make a definitive new conclusion that advances or clarified the current mechanistic status of the field. In particular, in budding yeast and other model organisms, it is clear that the factors that activate MCM complexes, converting them to active bidirectional helicases and causing origin firing, are present in limiting amounts relative to loaded MCM complexes (Mantiero (Zegerman), EMBO J 2011; Tanaka (Araki) Current Biology, 2011). Over-expression of these factors causes late regions to replicate earlier and leads to faster cell cycles and viability (in yeast) or developmental defects in metazoans (e.g. Collart (Zegerman) Science 2013). In yeast, where precise S-phase origin activation times can be assigned to individual origins, origins that are normally activated late in S-phase are now activated earlier. Thus limiting concentrations of MCM complex activation factors relative to the numbers of loaded MCM complexes can, to a substantial extent, explain origin activation time and the replication timing “program” of yeast chromosomes. In further support of this model early firing of centromere proximal origins is achieved by the ability of a kintechore complex, Ctf19, to recruit Dbf4, the regulatory subunit of Cdc7, a kinase, and a one of the limiting S-phase origin activation factors (Natsume Mol Cell 2013). The MCM complex is a direct substrate of this kinase, this its recruitment causes an origin to fire. In the absence of this recruitment mechanism, replication of yeast centromeric regions is specifically delayed. Another group of early origins use the forkhead transcription factors to recruit Dbf4 and activate the MCM complex (Fang, (Pasero, Lou) Genes Dev 2017). Thus it is clear that limiting S-phase factors can explain the stochastic behavior of origins to a large extent with early origins acting early because they possess mechanisms that increase the probability that they can recruit limiting S-phase factors).

Obviously, because MCM complex is the substrate for limiting S-phase activation factors, loaded MCM complexes must be present. The authors are correct in the assertion that fewer studies have addressed the level of MCM complex loading and, in theory, later replicating regions could result from either 1, a much lower abundance of MCM complexes or a lack of them all together or 2, the presence of MCM complexes that are considerably less competitive for recruiting the limiting S-phase factors. The question is whether the distribution of MCM complexes versus the accessibility of these loaded complexes to S-phase activation factors is the major determinant of replication timing patterns. The extant data, particularly from yeast, is that distribution of S-phase factors are a primary determinant. Thus, based on the title and the abstract of the current study, the authors conclusion that MCM complex abundance is the primary determinant of the replication timing program runs counter to the current view. Perhaps their use of a novel method to map MCM complexes is revealing something that has been missed by MCM ChIPSeq approaches, the primary method used to date to map licensed origins in vivo. Here are three detailed concerns:

1, In yeast, where precise efficiency values (Origin Efficiency Metric) and precise origin replication times have been mapped by a multitude of approaches, there is compelling evidence that that altering the concentrations of Dbf4 through discrete protein-protein interaction mechanisms, can substantially alter the activation timing of individual origins and thus the replication timing “program”. In addition, few studies have generated c evidence for a strong correlation between the levels of MCM complex loading (measured by MCM ChIPSeq) and the time at which an origin is activated. In the one published study that reported a correlation between the level of MCM complex signals and timing, the correlation is actually quite weak and not predictive, clearly suggesting that other factors contribute to determining an origin’s replication time (Das (Rhind) Genome Res, 2015).

Given that the authors use a new approach to assess loaded MCM complexes, additional analyses of the S. cerevisiae data should be performed to produce a clear picture of how the data from the MCM CheC method used here compares to recently published MCM ChIPSeq data and lots of other strong published data about yeast origins. The field knows so much about yeast origins that it makes sense to challenge the MCM CheC method more deeply in this model. In particular, the published MCM ChIPSeq data sets do not show a strong, highly predictive correlation between origin activation time in S-phase and the level of MCM complex loading at an origin. And while a weak correlation is reported in one study, it also quite clear that some late origins have MCM ChIP Seq signals that are similar to those of early origins. Therefore, the authors should address:

How well do the MCM CheC signals measured at individual origins in this study correlate with measured origin activation times?

How well do the relative MCM CheC signals compare to the relative MCM ChIPSeq signals from published studies? In other words, is MCM CheC method allowing you to detect/assess/infer relative differences in MCM complex levels at yeast origins that are lost by MCM ChIPSeq data? And, if so, how do you know whether the lower signals at late origins in your MCM CheC experiments are the result of less MCM being loaded OR the result of reduced access of the underlying DNA to MNase cleavage?

Given the strong conclusion the authors want to make using a new less scrutinized method that seems to be producing conclusions that run counter to published thinking in the field, such scrutiny is advisable. This information could then help readers better interpret the MCM CheC signals in eukaryotic cell types where MCM ChIPSeq has been relatively uninformative or confusing.

2, To a certain extent, it is not at all unexpected that at some level the abundance of licensed origins (loaded MCM complexes) should correlate with the replication timing program, for the simple reason that the MCM complex is the substrate for the S-phase factors and is essential for initiation. Thus, a chromosomal region lacking MCM complexes, regardless of its ability to recruit S-phase activation factors, will replicate later than neighboring regions with an abundance of loaded MCM complexes by the laws of physics. Indeed, a study from the Struhl lab (Miotto, PNAS, 2016) concludes that replication timing in human cells is determined largely by the density of ORC sites and the stochastic activation of origins. Given that ORC’s role is to load MCMs it is likely that ORC density must correlate with MCM complex density. Perhaps the study would be improved by being more explicit about whether the authors are thinking about the activation time of individual origins (loaded MCM complexes) or the relative replication time of a chromosomal region (and hence the abundance of detectable MCM complexes over a defined region of the genome). The distinct point nature of yeast origins makes it challenging to compare the bases for replication timing programs in yeast and other organisms--the current terminology/language is limiting/

3, Finally in this study under consideration, the authors report clear exceptions to the major conclusion “rule” that they assert. For example, they conclude that the Xa and Xi chromosomes, though they replicate at different times during S-phase, show similar patterns and abundance of MCM CheC signals, and thus that MCM loading cannot explain their replication timing differences. This result is interesting, but it seems like a major “exception” given that heterochromatin is quite abundant in metazoans.

In summary, the effort to apply a new approach to mapping MCM complexes in eukaryotic cells is admirable and the data are very likely revealing new facets about MCM complex behavior on chromosomes and its relationship to origin activity. However, the assay was used across multiple cell types without sufficiently rigorous challenging in yeast, the best developed model for origins. Thus neither the method or any single conceptual issue was sufficiently delved into to contribute a clear solid advance to the field. (Mcm subunits are not equivalent to transcription factors in dynamics and the assay was developed with those factors in mind) Thus, despite the title, the major new conclusions that will alter or challenge the current thinking in the field are unclear.

**Have all data underlying the figures and results presented in the manuscript been provided?**

Reviewer #1: Yes

Reviewer #2: Yes

Reviewer #3: Yes

PLOS authors have the option to publish the peer review history of their article (what does this mean?). If published, this will include your full peer review and any attached files.

Reviewer #1: **Yes: **Kevin Brick

Reviewer #2: No

Reviewer #3: No

---

## [Decision Letter · Decision Letter 1]

7 Apr 2021

Dear Dr Bedalov,

Thank you very much for submitting your revised Research Article entitled 'Chromosomal Mcm2-7 distribution and the genome replication program in species from yeast to humans' to PLOS Genetics.

The manuscript was fully evaluated at the editorial level and by independent peer reviewers. The reviewers appreciated the attention to an important problem. Two of the reviewers were satisfied with the revisions, but one reviewer raised some substantial concerns about the current manuscript, and I (ML) also have remaining substantial concerns. Based on the reviews, we will not be able to accept this version of the manuscript, but we would be willing to review a much-revised version. We cannot, of course, promise publication at that time.

The reviews are included below and/or attached; my comments are immediately below. 

1.  Of great concern is the continued absence of methods information sufficient to evaluate the data, let alone repeat it. This ranges from the trivial (formula for pombe media, growth media used for mammalian cells) to the profound (complete absence of methods for MNase-alone controls). The standard for completeness of methods should be that sufficient information is included to allow a third party to accurately replicate the experiments performed. None of the current protocols are sufficiently described, and given that this is an online journal without word limits, there is no reason not to be complete. If anything, one should err on the side of loquaciousness. Please be aware that unless full methods are provided, it will not be possible to evaluate remaining scientific concerns (for example, about the validity of MNase-only controls).

2. Of similar concern is the continued unavailability of underlying data. I could not find a data availability statement, the data underlying many of the graphs and figures are not provided, and it appears that data for the MNase controls is absent. Please refer to the PLOS data availability guidance (journals.plos.org/plosgenetics/s/data-availability, see also below) for what is required. Briefly, numerical data underlyling every graph must be available. Many authors find it convenient to include this in a single Excel workbook with separate worksheets for each figure panel, but there are alternatives listed in the guidance. Again, if data are not available then it will not be possible to evaluate remaining scientific concerns.

3. Actual numerical values and units for graphs and figures. In my opinion, most of the graphs and heat maps in the current manuscript are no better than cartoons. Please include, in every graph, a Y axis with real numbers and units that are defined explicitly in either the methods (if a term such as "relative abundance" is used repeatedly) or in the figure legend. Please include in every heatmap a spectrum bar with real numbers and units. "Avoiding confusion" is not a valid reason to omit such essential information.

4. Minor requests. a) In Figure 7 and the accompanying supplementary figure, it is very difficult to discern the transition from red to blue dots that are meant to mark initiation and termination zones. Please indicate the location of each of these--perhaps differently colored vertical arrows on the X axis?--using criteria that can be defined. It is not at all clear , for example, that Mcm peaks really correspond to measured initiation zones. b) please carefully proof references for proper species and gene name italicization, title capitalization, etc.

If you decide to revise the manuscript for further consideration at PLOS Genetics, please aim to resubmit within the next 60 days, unless it will take extra time to address the concerns of the reviewers, in which case we would appreciate an expected resubmission date by email to plosgenetics@plos.org.

[LINK]

We are sorry that we cannot be more positive about your manuscript at this stage. Please do not hesitate to contact us if you have any concerns or questions.

Yours sincerely,

Michael Lichten, Ph.D.

Associate Editor

PLOS Genetics

Gregory P. Copenhaver

Editor-in-Chief

PLOS Genetics

Reviewer's Responses to Questions

**Comments to the Authors:**

Reviewer #1: Review is uploaded as an attachment.

Reviewer #2: The authors have addressed my comments in good-faith. I am now satisfied that the manuscript is suitable for publication.

Reviewer #3: The authors have done due diligence and responded thoroughly to the reviews.

**Have all data underlying the figures and results presented in the manuscript been provided?**

Reviewer #1: **No: **MNase controls are not included in the GEO.

Reviewer #2: Yes

Reviewer #3: Yes

PLOS authors have the option to publish the peer review history of their article (what does this mean?). If published, this will include your full peer review and any attached files.

Reviewer #1: No

Reviewer #2: No

Reviewer #3: No

---

## [Editor Report · Decision Letter 2]

30 Jun 2021

Dear Dr Bedalov,

Thank you very much for submitting your Research Article entitled 'Chromosomal Mcm2-7 distribution and the genome replication program in species from yeast to humans' to PLOS Genetics.

The manuscript was fully evaluated at the editorial level, and I believe that you have done an excellent job of addressing the previous concerns, so I see no point in sending the manuscript out for review again. However, we identified some minor concerns that we ask you address in a revised manuscript. This should be relatively easy to do.

First, thank you for meeting requests for full methods details and for supplying underlying data. There are just a few changes that need to be made:

1. Figure 3B, underlying data. I could not find the numerical data underlying this graph in source_files_for_Figure1_to_Figure7A.zip. Please include it.

2. Source files for figures. Thank you for including these, which I know must have taken a lot of work. These files should be renamed something like "S1_source_files_for_Figure1_to_Figure7A.zip" (to pick an example), "S2_source_files_for_Figure7B_part1_of_2.zip" etc. etc. (see journals.plos.org/plosgenetics/s/supporting-information for further information). Then, each Sx source file should be listed in the manuscript along with the other supplementary files (i.e. on page 49 and following of the current manuscript. For example:

Supplemental information:

S1_source_files_for_Figure1_to_Figure7A.zip  Zip file containing the following: data underlying Figure 1A, B, C, and code example; data underlying Figure 2A, B, C;....etc.

3. In the data availability statement, please add that numerical data and code underlying all figures can be found in supporting information files S1-S12. 

[LINK]

Yours sincerely,

Michael Lichten, Ph.D.

Associate Editor

PLOS Genetics

Gregory P. Copenhaver

Editor-in-Chief

PLOS Genetics

---

## [Editor Report · Decision Letter 3]

13 Jul 2021

Dear Dr Bedalov,

We are pleased to inform you that your manuscript entitled "Chromosomal Mcm2-7 distribution and the genome replication program in species from yeast to humans" has been editorially accepted for publication in PLOS Genetics. Congratulations!

Yours sincerely,

Michael Lichten, Ph.D.

Associate Editor

PLOS Genetics

Gregory P. Copenhaver

Editor-in-Chief

PLOS Genetics

Comments from the reviewers (if applicable):

**Data Deposition**

http://datadryad.org/submit?journalID=pgenetics&manu=PGENETICS-D-20-01578R3

**Press Queries**

---

## [Editor Report · Acceptance letter]

27 Aug 2021

PGENETICS-D-20-01578R3 

Chromosomal Mcm2-7 distribution and the genome replication program in species from yeast to humans 

Dear Dr Bedalov, 

We are pleased to inform you that your manuscript entitled "Chromosomal Mcm2-7 distribution and the genome replication program in species from yeast to humans" has been formally accepted for publication in PLOS Genetics! Your manuscript is now with our production department and you will be notified of the publication date in due course.

With kind regards,

Andrea Szabo

PLOS Genetics

On behalf of:
